# A Rotated Hyperbolic Wrapped Normal Distribution for Hierarchical Representation Learning

**Seunghyuk Cho**[1]     **Juyong Lee**[1]     **Jaesik Park**[1,2]     **Dongwoo Kim**[1,2]

**CSED POSTECH**[1]                          **GSAI POSTECH**[2]

## Abstract

We present a rotated hyperbolic wrapped normal distribution (RoWN), a simple yet effective alteration of a hyperbolic wrapped normal distribution (HWN). The HWN expands the domain of probabilistic modeling from Euclidean to hyperbolic space, where a tree can be embedded with arbitrary low distortion in theory. In this work, we analyze the geometric properties of the *diagonal* HWN, a standard choice of distribution in probabilistic modeling. The analysis shows that the distribution is inappropriate to represent the data points at the same hierarchy level through their angular distance with the same norm in the Poincaré disk model. We then empirically verify the presence of limitations of HWN, and show how RoWN, the proposed distribution, can alleviate the limitations on various hierarchical datasets, including noisy synthetic binary tree, WordNet, and Atari 2600 Breakout. The code is available at `https://github.com/ml-postech/RoWN`.

## 1   Introduction

Hyperbolic space has served as an effective medium to learn parsimonious representations of hierarchical data, including vocabulary with relationships (Nickel and Kiela, 2017, 2018; Tifrea et al., 2019), knowledge graphs (Chami et al., 2020; Sun et al., 2020), and social networks (Zhao et al., 2011; Shavitt and Tankel, 2008). Recent studies reveal that the underlying anatomy in much complex data is non-Euclidean, supporting the success of representation learning in hyperbolic space (Bronstein et al., 2017). However, due to the absence of well-defined distribution that is easy to sample and has an analytic density function in hyperbolic space, earlier studies have focused on the non-probabilistic learning framework.

Hyperbolic wrapped normal distribution (HWN) has been recently proposed as an alternative to the celebrated normal distribution in Euclidean space. With an analytic density function and easiness of sampling, the HWN expands the domain of probabilistic modeling from Euclidean to hyperbolic space. The HWN has been successfully applied in various probabilistic models, including VAE (Kingma and Welling, 2014) and probabilistic word embeddings (Vilnis and McCallum, 2015). However, unlike the normal distribution in Euclidean space, the geometric characteristics of the HWN have not been fully understood so far. Therefore, figuring out what can be done or not with the HWN is difficult.

In this work, we analyze the geometric properties of the *diagonal* HWN, a standard choice of distribution in many probabilistic models (Mathieu et al., 2019; Nagano et al., 2019). Based on the observation that the principal axes of the diagonal normal distribution in Euclidean space have a parallel structure with the standard bases, we also focus on the structure of the principal axes of HWN in hyperbolic space to delve into a deeper understanding. Our analysis shows that the principal axes of the diagonal HWN are locally parallel to the standard bases in the Poincaré disk model.

Figure 1 (a) shows a common understanding on learned representation of hierarchical data in the Poincaré disk. The hierarchical structure spreads out like the spokes of a wheel. The local variation

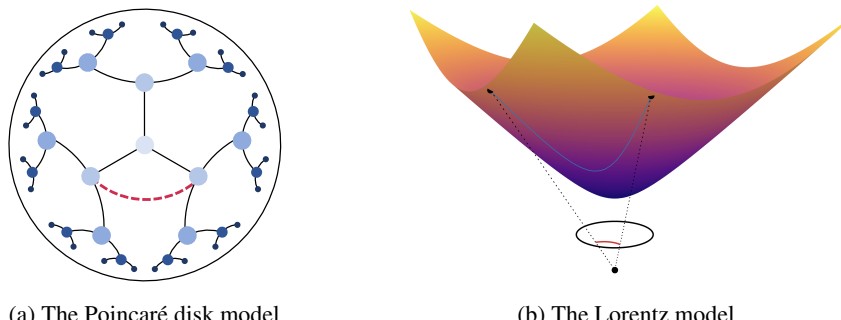

(a) The Poincaré disk model

(b) The Lorentz model

Figure 1: Visualization of hyperbolic space models. (a) The Poincaré disk model can embed a given tree-structured data with low distortions as shown in the illustration. The black segments refer to the shortest path between the points in the Poincaré disk model. The red dashed line denotes the continuous points in the same hierarchy level. (b) The Lorentz model is another model for hyperbolic space. The blue line is the shortest path between the points on the Lorentz model. The red line is the projected geodesic via the diffeomorphism, which is still a geodesic of the Poincaré disk model.

in a hierarchy can then be represented as an angular difference between nodes at the same level of the norm, i.e., the principal axis representing local variation is orthogonal to the radial axis. However, our analysis of the geometric property reveals that the local variation can only be modeled along with the standard bases with the diagonal HWN.

To fix the structure of the principal axes in the diagonal HWN, we propose a simple yet effective alteration of HWN, named a rotated hyperbolic wrapped normal distribution (RoWN). By rotating the diagonal covariance matrix before parallel transportation of HWN, we could resolve the limitations in local variation structure while keeping the valuable properties, such as easy sampling and tractable density of the original HWN.

We verify the representation learned with RoWN agrees with the common characteristic of representation in hyperbolic space, which is barely observable with the diagonal and the full covariance HWN, by using *synthetic noisy binary tree* dataset. We demonstrate the usefulness of RoWN on the benchmark datasets: WordNet and Atari 2600 breakout. We summarize our contributions as follows:

1. We provide an analysis of the geometric properties of HWN and its potential limitations in representation learning.

2. We propose a novel and efficient method of using hyperbolic distribution, namely RoWN, and apply it to probabilistic models.

3. We demonstrate the performance of RoWN through the comparison with the Euclidean normal distribution, diagonal covariance HWN, and full covariance HWN on one synthetic dataset and two benchmark datasets.

## 2 Preliminaries

This section reviews the wrapped normal distribution defined on the Lorentz model. We first introduce the Lorentz model of hyperbolic space and necessary concepts to understand the wrapped normal distribution.

### 2.1 The Lorentz model

Hyperbolic space is a non-Euclidean space, having a constant negative Gaussian curvature. Figure 1 illustrates two models for hyperbolic space, among four standard equivalent models: (1) Klein model, (2) the Poincaré disk model, (3) the Lorentz (hyperboloid) model, and (4) the Poincaré half-plane model. In particular, the Lorentz model is famous for its numerical stability in computing the distance and comes with a simpler closed form of geodesics (Nickel and Kiela, 2018). The Lorentz model $\mathbb{L}^n$ is the Riemannian manifold consisting of the set of points $z \in \mathbb{R}^{n+1}$ satisfying $\langle z, z \rangle_{\mathcal{L}} = -1$ and

$z_0 > 0$, where the Lorentizan product $\langle \cdot, \cdot \rangle_{\mathcal{L}}$ is defined as:

$$\langle \boldsymbol{x}, \boldsymbol{y} \rangle_{\mathcal{L}} := -x_0 y_0 + \sum_{i=1}^{n} x_i y_i,$$

which also works as the metric tensor on hyperbolic space, i.e., the metric tensor $g$ of the Lorentz model is $g(\boldsymbol{x}) = \mathrm{diag}[-1, 1, \cdots, 1]$

## 2.2 Tangent space of the Lorentz model

We denote the tangent space of $\boldsymbol{x} \in \mathbb{L}^n$ as $\mathcal{T}_{\boldsymbol{x}} \mathbb{L}^n$, which is a set of points satisfying the orthogonality relation with $\boldsymbol{x}$ in terms of the Lorentzian product: $T_{\boldsymbol{x}} \mathbb{L}^n := \{\boldsymbol{u} : \langle \boldsymbol{u}, \boldsymbol{x} \rangle_{\mathcal{L}} = 0\}$. The metric tensor $g$ induces an inner product of two tangent vectors from a tangent space. Geodesic $\gamma : [0, 1] \rightarrow \mathbb{L}^n$ generalizes straight lines in the Riemannian manifold which is the shortest curve between two points. The exponential map $\exp_{\boldsymbol{x}} : \mathcal{T}_{\boldsymbol{x}} \mathbb{L}^n \rightarrow \mathbb{L}^n$ maps a tangent vector $\boldsymbol{u} \in \mathcal{T}_{\boldsymbol{x}} \mathbb{L}^n$ onto $\mathbb{L}^n$ as:

$$\exp_{\boldsymbol{x}}(\boldsymbol{u}) := \cosh(\|\boldsymbol{u}\|_{\mathcal{L}})\boldsymbol{x} + \sinh(\|\boldsymbol{u}\|_{\mathcal{L}})\frac{\boldsymbol{u}}{\|\boldsymbol{u}\|_{\mathcal{L}}}, \tag{1}$$

such that $\exp_{\boldsymbol{x}}(\boldsymbol{u}) = \boldsymbol{y}, \gamma(0) = \boldsymbol{x}, \gamma(1) = \boldsymbol{y}$. The log map, inverse of the exponential map, is defined as $\log_{\boldsymbol{x}}(\boldsymbol{y}) := \exp_{\boldsymbol{x}}^{-1}(\boldsymbol{u})$.

Parallel transport is an operation that transports a tangent vector in the tangent space at $\boldsymbol{x}$ to another vector in the tangent space at $\boldsymbol{y}$ along the geodesic from $\boldsymbol{x}$ to $\boldsymbol{y}$ without losing the parallel property. The parallel transport in the Lorentz model is given by:

$$\mathrm{PT}_{\boldsymbol{x} \rightarrow \boldsymbol{y}}(\boldsymbol{v}) := \boldsymbol{v} + \frac{\langle \boldsymbol{y} - \alpha \boldsymbol{x}, \boldsymbol{v} \rangle_{\mathcal{L}}}{\alpha + 1}(\boldsymbol{x} + \boldsymbol{y}), \tag{2}$$

where $\alpha = -\langle \boldsymbol{x}, \boldsymbol{y} \rangle_{\mathcal{L}}$.

## 2.3 Hyperbolic wrapped normal distribution

One of the key challenges in adopting hyperbolic space to probabilistic models is finding a distribution on hyperbolic space that is easy to sample and has a closed-form density function. The two most common distributions of hyperbolic space used in previous work are Riemannian normal distribution (Mathieu et al., 2019; Said et al., 2014) and hyperbolic wrapped normal distribution (HWN) (Nagano et al., 2019). Our work is mainly based on the HWN because sampling in Riemannian normal distribution is limited with only a unit variance.

The sampling process with the HWN follows:

1. Sample $\boldsymbol{v} \in \mathbb{R}^n$ from a Gaussian distribution $\mathcal{N}(\boldsymbol{0}, \Sigma)$ in Euclidean space.

2. Parallel transport the vector $[0, \boldsymbol{v}] \in \mathcal{T}_{\boldsymbol{0}_{\mathcal{L}}} \mathbb{L}^n$ to the tangent space.

3. Project the transported tangent vector to $\mathbb{L}^n$ using exponential mapping.

The density of a sample can be measured via the change of variable method. For convenience, we denote the procedures 2 and 3 as a single operation:

$$f_{\boldsymbol{\mu}}(\boldsymbol{v}) := \exp_{\boldsymbol{\mu}}(\mathrm{PT}_{\boldsymbol{0}_{\mathcal{L}} \rightarrow \boldsymbol{\mu}}([0, \boldsymbol{v}])), \tag{3}$$

with given $\boldsymbol{\mu} \in \mathbb{L}^n$ and $\boldsymbol{v} \in \mathbb{R}^n$, and $\boldsymbol{0}_{\mathcal{L}} = [1, \ldots, 0]$ stands for the origin of $\mathbb{L}^n$.

# 3 Rotated Hyperbolic Wrapped Normal Distribution

This section introduces several observations on the geometric properties of the HWN distribution transformation from the Euclidean space to hyperbolic space. We show the limitations of the diagonal HWN for representation learning of hierarchical data. Then, we propose a simple yet effective modification of the HWN, called a rotated hyperbolic wrapped normal distribution.

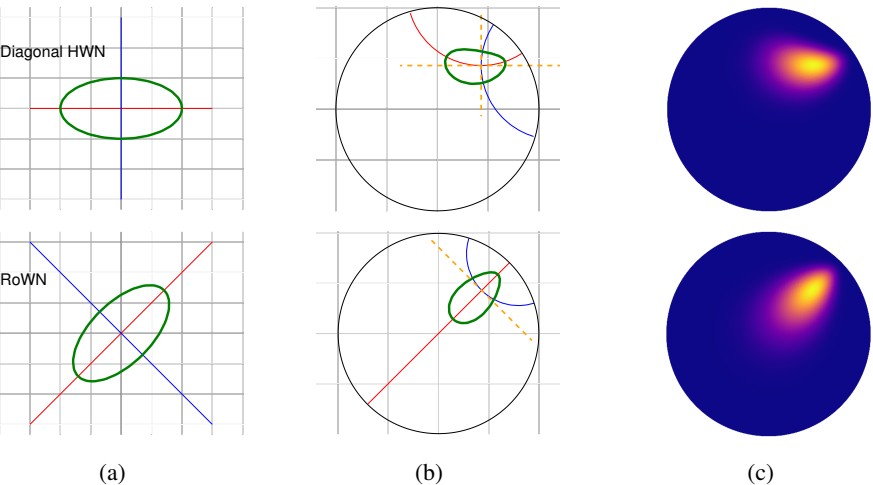

(a)                          (b)                          (c)

Figure 2: Visualization of (a) the principal axes of the normal distribution in the Euclidean space, (b) the transported version of two axes in hyperbolic space, and (c) the probability density plot of the distributions. (a) The contour line (green) can be represented as an ellipse with major principal axis (red) and minor principal axis (blue). (b) The transformed principal axes become geodesics in hyperbolic space. The major principal axis of the RoWN passes the hyperbolic origin and crosses with the minor axis at the mean point, whereas the major and minor axes of the diagonal HWN is locally parallel to the standard axes. (c) The principal axes determine the shapes of the variational distributions.

## 3.1 Observations on the hyperbolic wrapped normal distribution

First, we investigate the changes in the normal distribution during the transformation from Euclidean to hyperbolic space by Equation 3. As the principal axes characterize the covariance structure of the normal distribution in the Euclidean space, we investigate how the principal axes are transformed in hyperbolic space. Before deriving our main proposition, we first show that the straight lines that pass through the origin in the Euclidean space are transformed to the geodesics in hyperbolic space by Equation 3.

**Proposition 1.** *Suppose $\ell_{\boldsymbol{s}}(t) = t\boldsymbol{s} \in \mathbb{R}^n$ be a line passing through the origin, where $\boldsymbol{s} \in \mathbb{R}^n$ is a directional vector. Then the curve $f_{\boldsymbol{\mu}}(\ell_{\boldsymbol{s}}(t))$ in the Lorentz model $\mathbb{L}^n$ becomes a geodesic.*

The proof of the proposition is provided in Appendix A. The proposition indicates that every straight line that passes through the origin including the principal axes, is transformed into a geodesic in hyperbolic space.

Based on the first proposition, we provide our main proposition, which fully characterizes the structure of principal axes when projected to the Poincaré disk model:

**Proposition 2.** *Define $\mathrm{Proj}(\boldsymbol{u})$ to be the projection function from the Lorentz model to the Poincaré model, i.e., $\mathrm{Proj}(\boldsymbol{u}) = \frac{x_{1:}(\boldsymbol{u})}{x_0(\boldsymbol{u})+1}, \forall \boldsymbol{u} \in \mathbb{L}^n$. If $\ell_{\boldsymbol{s}}$ is a principal axis of the normal distribution defined in $\mathbb{R}^n$ and $\boldsymbol{\mu}$ the mean of HWN in $\mathbb{L}^n$, then $\boldsymbol{s}$ is the tangent vector of $\mathrm{Proj}(f_{\boldsymbol{\mu}}(\ell_{\boldsymbol{s}}))$ on $\mathrm{Proj}(\boldsymbol{\mu})$.*

The proof of the proposition is provided in Appendix B. The proposition reveals that the principal axes of the HWN are locally parallel to the standard bases in the Poincaré disk model. To visualize the proposition, we plot the contour line and the principal axes of the two-dimensional diagonal normal distribution before and after the transformation in Figure 2. We observe that the tangent lines of the transformed principal axes are parallel to the standard bases in hyperbolic space.

When a popular diagonal normal distribution is employed as a variational distribution, the locally parallel principal axes might be problematic in learning hierarchical representations. For example, suppose one tries to represent the variability along the radial direction or in angular differences. In the case, both the major (red) and minor (blue) axes in Figure 2b cannot model the variability properly.

**Algorithm 1** Sampling process with the rotated hyperbolic wrapped normal distribution

---

**Input** Mean $\boldsymbol{\mu} \in \mathbb{L}^n$, diagonal covariance matrix $\Sigma \in \mathbb{R}^{n \times n}$

**Output** Sample $\boldsymbol{z} \in \mathbb{L}^n$

1: $\boldsymbol{x} = [\pm 1, \ldots, 0] \in \mathbb{R}^n, \boldsymbol{y} = \boldsymbol{\mu}_{1:}/\|\boldsymbol{\mu}_{1:}\|$          $\triangleright \pm$ is determined by the sign of $\boldsymbol{\mu}_0$

2: $\boldsymbol{R} = \boldsymbol{I} + (\boldsymbol{y}^T\boldsymbol{x} - \boldsymbol{x}^T\boldsymbol{y}) + (\boldsymbol{y}^T\boldsymbol{x} - \boldsymbol{x}^T\boldsymbol{y})^2/(1 + \langle \boldsymbol{x}, \boldsymbol{y} \rangle)$

3: Rotate $\hat{\Sigma} = \boldsymbol{R}\Sigma\boldsymbol{R}^T$

4: Sample $\boldsymbol{v} \sim \mathcal{N}(\boldsymbol{0}, \hat{\Sigma})$

5: **return** $\boldsymbol{z} = f_{\boldsymbol{\mu}}(\boldsymbol{v})$

---

## 3.2 Rotated hyperbolic wrapped normal distribution

Based on the observation, we propose a simple yet effective alternative to the diagonal HWN, a rotated hyperbolic wrapped normal distribution (RoWN). Rotating the covariance matrix to the direction of $\boldsymbol{\mu}$ enables aligning the major axis of the normal distribution in the Euclidean space to the radial axis in hyperbolic space as visualized in the Figure 2.

To construct the distribution, we start with a mean vector $\boldsymbol{\mu} \in \mathbb{L}^n$ and a diagonal covariance matrix $\Sigma$ as in the standard HWN. We change the covariance matrix of the normal distribution as follows:

1. Compute the rotation matrix $\boldsymbol{R}$ that rotates the x-axis ($[\pm 1, \ldots, 0] \in \mathbb{R}^n$) to $\boldsymbol{\mu}_{1:}$.

2. Substitute the covariance matrix of Gaussian normal with $\boldsymbol{R}\Sigma\boldsymbol{R}^T$.

Thus, the rotation matrix $\boldsymbol{R}$, which rotates a unit vector from $\boldsymbol{x}$ to $\boldsymbol{y}$, can be computed as:

$$\boldsymbol{R} = \boldsymbol{I} + (\boldsymbol{y}^T\boldsymbol{x} - \boldsymbol{x}^T\boldsymbol{y}) + \frac{1}{1 + \langle \boldsymbol{x}, \boldsymbol{y} \rangle}(\boldsymbol{y}^T\boldsymbol{x} - \boldsymbol{x}^T\boldsymbol{y})^2. \tag{4}$$

The pseudo-code of the sampling process of RoWN is in Algorithm 1. Note that the construction is straightforward but still keeps the following benefits of the HWN: 1) The sampling can be done efficiently, and 2) the computation of the probability density of the samples is tractable. As the HWN provides a tractable probability density function for any kind of covariance matrix, we can easily compute the probability density of a given sample from RoWN. See Appendix C for more details.

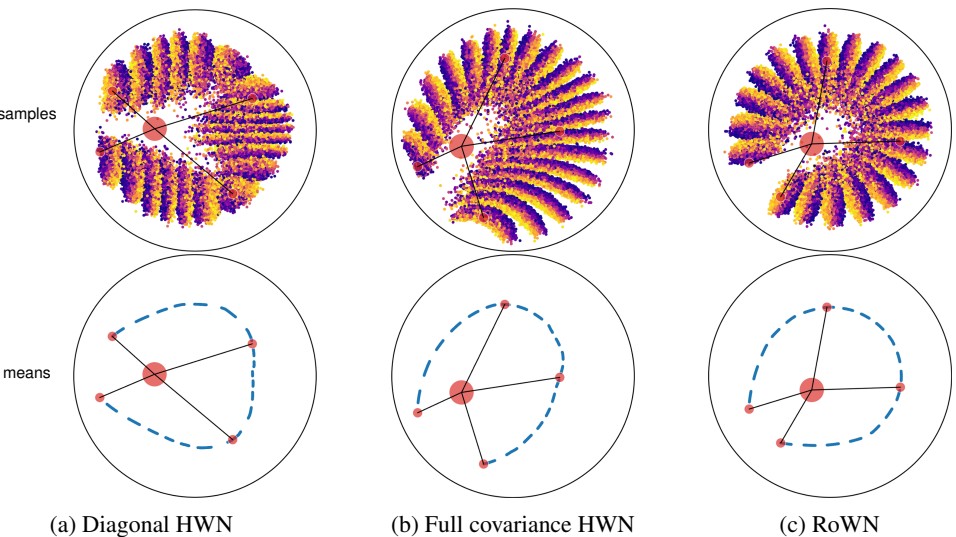

     (a) Diagonal HWN          (b) Full covariance HWN        (c) RoWN

Figure 3: Visualization of the variational distribution of hyperbolic VAE on a synthetic binary tree dataset with different variational distributions. The red dots denote the variational means of the root and four representative children. The upper row shows the samples from the variational distributions where the color denotes the level of noise. The bottom row shows the means of the variational distributions. Overall, RoWN better aligns local variation in angular difference.

We empirically demonstrate the influence of different variational distributions in Figure 3. We visualize the variational distributions of the synthetic binary tree dataset of depth two after training a hyperbolic VAE. As the result shows, the model with the diagonal HWN represents the variation in a child parallel to the standard bases, whereas the model with RoWN represents the variation in angular difference. Please check the detailed description on the synthetic dataset in Section 4.2.

## 4 Experiments

In this section, we first explain the two applications of the distribution defined on hyperbolic space: hyperbolic VAE and probabilistic hyperbolic word embedding model. We then conduct three different experiments to compare the performance of RoWN with four baselines, including the Gaussian distribution in the Euclidean space, the isotropic HWN (Nagano et al., 2019), the diagonal HWN, and the full covariance HWN. We also provide an additional study on a variant of RoWN with learnable rotation direction $\boldsymbol{y}$ in Algorithm 1, and the results are in Table 9. The details of the experiments are described in Appendix D.

### 4.1 Applications of the hyperbolic distribution

**Hyperbolic VAE.** The hyperbolic VAE, whose latent space is hyperbolic space, has been shown to be efficient for capturing the hierarchical structure of the data (Nagano et al., 2019; Mathieu et al., 2019). The evidence lower bound of the hyperbolic VAE can be written as:

$$\mathcal{L}_{\mathrm{ELBO}}(\theta, \phi) := \mathbb{E}_{q_\phi(\boldsymbol{z}|\boldsymbol{x}) \cdot \sqrt{\det(g)}}[\log p_\theta(\boldsymbol{x} \mid \boldsymbol{z})] - D_{\mathrm{KL}}\left(q_\phi(\boldsymbol{x} \mid \boldsymbol{z}) \cdot \sqrt{\det(g)} \parallel p(\boldsymbol{z}) \cdot \sqrt{\det(g)}\right),$$

where $q_\phi$ is the encoder, $p_\theta$ is the decoder, $g$ is the metric tensor of the chosen model of hyperbolic space, and $p(\boldsymbol{z})$ is a prior distribution. The distributions defined on hyperbolic spaces, such as HWN and RoWN, are used to define encoder $q_\phi(\boldsymbol{z} \mid \boldsymbol{x}) \cdot \sqrt{\det(g)}$ and prior $p(\boldsymbol{z}) \cdot \sqrt{\det(g)}$. In hyperbolic VAEs, due to the absence of the closed-form KL divergence in the hyperbolic distributions, the KL divergence between the encoder and the prior is usually approximated with Monte-Carlo sampling (Nagano et al., 2019; Mathieu et al., 2019).

**Probabilistic hyperbolic word embedding model.** The probabilistic word embedding models aim to learn probabilistic representations of words (Vilnis and McCallum, 2015; Nagano et al., 2019; Tifrea et al., 2019). The embeddings learned on hyperbolic spaces have shown better performance than the Euclidean counterpart (Nagano et al., 2019; Tifrea et al., 2019). Given the hypernymy relationships between the words, the probabilistic hyperbolic embedding model learns the probabilistic representation of words by minimizing the following objective:

$$\mathcal{L}_{\mathrm{word}}(\boldsymbol{\theta}) := \mathbb{E}_{(s \sim t, s \not\sim t')}[\max(0, m + D_{\mathrm{KL}}(q_s \parallel q_t) - D_{\mathrm{KL}}(q_s \parallel q_{t'}))], \tag{5}$$

where $m$ is a margin, $q_i$ is a distribution for word $i$ parameterized by $\theta$, and $s \sim t$ and $s \not\sim t$ denote the presence and absence of hypernymy relation between word pair $s$ and $t$, respectively.

### 4.2 Noisy synthetic binary tree

A synthetic binary tree dataset is first used to show the performance of representing hierarchy in Nagano et al. (2019), where each node in a tree corresponds to a sequence of binary values. Figure 4a shows an example of the depth three binary tree, where a parent and child only differ in one digit. We add spherical noises to the nodes in the same level of hierarchy as described in Figure 4a as the noisy samples. With the noisy samples, we can create a dataset containing a local-level variation in the hierarchy. For the experiments, we uniformly sample the spherical noise from $[0, \pi/4]$.

We train hyperbolic VAE on *noisy synthetic binary tree* with varying depths. The detailed model description is available in Appendix D.1. We set the latent dimension the same as the depth. We report 1) the correlation between the hamming distance and the embedding distance and 2) the correlation between the depth and the Poincaré norm of the embeddings. The first correlation is computed over all possible pairs of test points. As Table 1 shows, the full covariance HWN and RoWN improve the diagonal HWN except depth six, outperforming the Euclidean model in every setting. RoWN preserves the depth information better than the other distributions in general. We additionally visualize the variational mean obtained by training the tree of depth three in Figure 4b, where the hierarchical structure is well preserved in the hyperbolic embedding space.

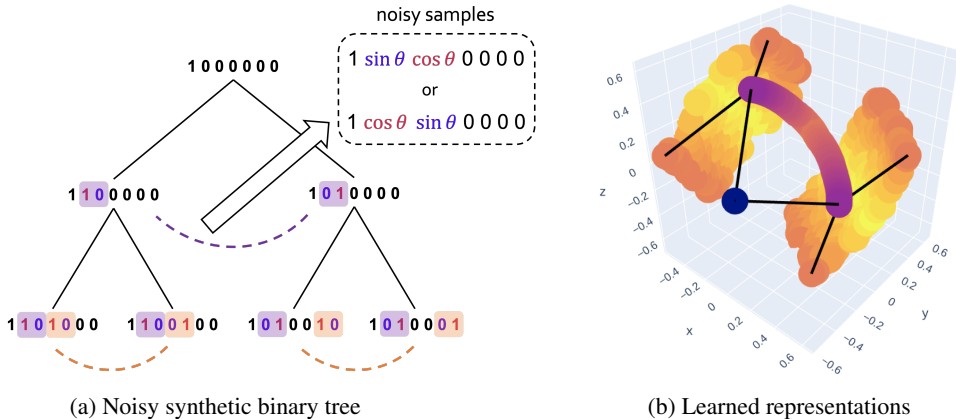

(a) Noisy synthetic binary tree

(b) Learned representations

Figure 4: Illustration of a noisy synthetic binary tree. (a) We construct a noisy synthetic binary tree by adding spherical noises defined with $\theta$. The continuous samples are generated at the same distance from the root. (b) We train the depth three noisy synthetic binary tree with hyperbolic VAE with RoWN as a variational distribution and visualize the means of the variational distributions. The black lines show the underlying hierarchical structure, and the color denotes the level of noise. The hierarchy and the local variations are well preserved in the representations.

## 4.3 WordNet

We train a probabilistic word embedding model with WordNet dataset (Fellbaum, 1998), which consists of 82,115 nouns and 743,241 hypernymy relationships. We have initialized the embeddings from $\mathcal{N}(0, 0.01I)$, which are then moved to the Lorentz model using the exponential map. We use the learning rate warm-up proposed in (Nagano et al., 2019). We evaluate the learned representations by computing the average rank of all the hypernymies. The rank of a given pair of words $s$ and $t$ is computed among the distances between all possible pairs of the words $s$ and $t'$ without hypernymy. Table 2 shows the empirical performances of representing the word data. We report the performance with the mean rank (MR) and the mean average precision (mAP). RoWN preserves the hierarchical structure better than the other distributions, while the full covariance HWN often performs worse than RoWN.

**Failure of the full covariance HWN.** In theory, the full covariance HWN needs to have at least a similar performance to RoWN since the RoWN is the particular case of the full covariance. The

Table 1: Results of *noisy synthetic binary tree*. The results are averaged over 10 runs. The hyperbolic models outperform the Euclidean model in all settings. Overall, RoWN preserves the hierarchical information better than the other distributions.

|  |  | Depth | | | |
|---|---|---|---|---|---|
|  |  | 4 | 5 | 6 | 7 |
| Correlation w/ distance | Euclidean | $0.748_{\pm.032}$ | $0.740_{\pm.013}$ | $0.741_{\pm.008}$ | $0.733_{\pm.014}$ |
|  | HWN (isotropic $\Sigma$) | $0.773_{\pm.030}$ | $0.809_{\pm.016}$ | $0.798_{\pm.008}$ | $0.735_{\pm.022}$ |
|  | HWN (diagonal $\Sigma$) | $0.814_{\pm.008}$ | $0.791_{\pm.023}$ | $0.817_{\pm.010}$ | $0.759_{\pm.025}$ |
|  | HWN (full $\Sigma$) | $0.827_{\pm.015}$ | $0.798_{\pm.026}$ | $0.798_{\pm.010}$ | $0.794_{\pm.014}$ |
|  | RoWN | $0.820_{\pm.015}$ | $0.807_{\pm.017}$ | $0.822_{\pm.017}$ | $0.788_{\pm.016}$ |
| Correlation w/ depth | Euclidean | $0.762_{\pm.117}$ | $0.807_{\pm.038}$ | $0.712_{\pm.054}$ | $0.612_{\pm.049}$ |
|  | HWN (isotropic $\Sigma$) | $0.902_{\pm.033}$ | $0.867_{\pm.034}$ | $0.811_{\pm.029}$ | $0.602_{\pm.066}$ |
|  | HWN (diagonal $\Sigma$) | $0.918_{\pm.028}$ | $0.808_{\pm.076}$ | $0.862_{\pm.035}$ | $0.697_{\pm.076}$ |
|  | HWN (full $\Sigma$) | $0.956_{\pm.015}$ | $0.878_{\pm.051}$ | $0.870_{\pm.033}$ | $0.815_{\pm.055}$ |
|  | RoWN | $0.930_{\pm.026}$ | $0.911_{\pm.027}$ | $0.901_{\pm.034}$ | $0.827_{\pm.047}$ |

Table 2: Results of *WordNet*. The results are an average of 5 runs. Based on the rank of hypernymy pairs among non-hypernymy pairs, we report the mean rank (MR) and mean average precision (mAP) for evaluation.

|  |  | Latent dimension | | |
|---|---|---|---|---|
|  |  | 5 | 10 | 20 |
| MR | Euclidean | $13.968_{\pm 0.504}$ | $3.862_{\pm 0.281}$ | $1.955_{\pm 0.157}$ |
|  | HWN (isotropic $\Sigma$) | $14.568_{\pm 2.203}$ | $4.470_{\pm 0.669}$ | $3.125_{\pm 0.455}$ |
|  | HWN (diagonal $\Sigma$) | $16.590_{\pm 1.146}$ | $3.891_{\pm 0.447}$ | $2.062_{\pm 0.088}$ |
|  | HWN (full $\Sigma$) | $557.309_{\pm 18.006}$ | $466.513_{\pm 75.142}$ | $599.140_{\pm 18.916}$ |
|  | RoWN | $16.271_{\pm 2.985}$ | $2.888_{\pm 0.162}$ | $1.783_{\pm 0.090}$ |
| mAP | Euclidean | $0.565_{\pm 0.014}$ | $0.801_{\pm 0.020}$ | $0.902_{\pm 0.008}$ |
|  | HWN (isotropic $\Sigma$) | $0.617_{\pm 0.012}$ | $0.820_{\pm 0.013}$ | $0.847_{\pm 0.017}$ |
|  | HWN (diagonal $\Sigma$) | $0.565_{\pm 0.020}$ | $0.805_{\pm 0.015}$ | $0.905_{\pm 0.007}$ |
|  | HWN (full $\Sigma$) | $0.032_{\pm 0.003}$ | $0.063_{\pm 0.005}$ | $0.079_{\pm 0.021}$ |
|  | RoWN | $0.593_{\pm 0.024}$ | $0.844_{\pm 0.009}$ | $0.921_{\pm 0.005}$ |

performance of the full covariance HWN often performs worse with the probabilistic hyperbolic word embedding models than the hyperbolic VAEs. We speculate that reason is because of the relatively simple prior in the hyperbolic VAE, whereas the probabilistic word embedding models need to compute the KL divergence between the full covariance HWNs. Our additional experiments reported in Table 11 confirm our speculation as the number of training samples for the KL divergence increases, the performance increases slightly.

**Discussion of the root placement.**    Note that due to the isometry of the geometry, the root node can be placed anywhere in hyperbolic space. In other words, infinitely many sets of embeddings preserve the same pairwise distances between the nodes. To place the root near the origin, we initialize all embeddings with the zero mean distribution as done in (Nagano et al., 2019). We find that this initialization helps the root placed near the origin. A study on the effects of the initialization methods is shown in Table 3. Further details about the root placement can be found in Appendix E.

## 4.4    Atari 2600 Breakout

Trajectories of some Atari 2600 games can be structured as a tree-like hierarchy along the time horizon as Nagano et al. (2019) points out. For example, given Atari 2600 Breakout, the root node can be the starting state of the game where no blocks have been broken. As the game progresses the states of the blocks form a hierarchical structure depending on which blocks have been broken. To

Table 3: The effects of initializations. We test different initializations on the deterministic hyperbolic word embedding models trained with the subset of the WordNet dataset. The `near zero vector` model initializes the embeddings with uniform distribution $\mathcal{U}(-0.001, 0.001)$, while the `near one vector` model initializes the embeddings with uniform distribution $\mathcal{U}(0.999, 1.001)$. While the Poincaré norm of the root node differs, the other metrics remain similar.

|  |  | Latent dimension | | | |
|---|---|---|---|---|---|
|  |  | 2 | 5 | 10 | 20 |
| MR | Near zero vector | $4.346_{\pm .643}$ | $3.270_{\pm .144}$ | $2.828_{\pm .098}$ | $2.508_{\pm .049}$ |
|  | Near one vector | $4.029_{\pm .415}$ | $3.209_{\pm .094}$ | $2.856_{\pm .075}$ | $2.491_{\pm .051}$ |
| mAP | Near zero vector | $0.821_{\pm .016}$ | $0.891_{\pm .006}$ | $0.891_{\pm .005}$ | $0.895_{\pm .003}$ |
|  | Near one vector | $0.829_{\pm .010}$ | $0.894_{\pm .004}$ | $0.891_{\pm .004}$ | $0.896_{\pm .003}$ |
| The Poincaré norm of the root node | Near zero vector | $0.122_{\pm .036}$ | $0.042_{\pm .015}$ | $0.031_{\pm .007}$ | $0.024_{\pm .004}$ |
|  | Near one vector | $0.505_{\pm .076}$ | $0.650_{\pm .010}$ | $0.732_{\pm .003}$ | $0.801_{\pm .001}$ |

Table 4: Results of *Atari 2600 Breakout*. The results are averaged over 5 runs. We measure the correlation between the score of an image and the Poincaré norm of the variational mean. The HWN with the isotropic covariance can be viewed as a variant of RoWN.

| | | Latent dimension | | |
|---|---|---|---|---|
| | | 10 | 15 | 20 |
| Correlation btw. score and norm | Euclidean | $0.379_{\pm.007}$ | $0.436_{\pm.029}$ | $0.479_{\pm.020}$ |
| | HWN (isotropic $\Sigma$) | $0.513_{\pm.012}$ | $0.598_{\pm.021}$ | $0.607_{\pm.015}$ |
| | HWN (diagonal $\Sigma$) | $0.478_{\pm.011}$ | $0.513_{\pm.006}$ | $0.513_{\pm.008}$ |
| | HWN (full $\Sigma$) | $0.483_{\pm.011}$ | $0.520_{\pm.009}$ | $0.563_{\pm.010}$ |
| | RoWN | $0.497_{\pm.014}$ | $0.556_{\pm.014}$ | $0.561_{\pm.029}$ |

learn the implicit hierarchy that can be observed from the trajectories of Breakout, we train the VAE models with the Atari 2600 Breakout images.

The images of Breakout are collected by using a pre-trained Deep Q-network (Mnih et al., 2015) and divided into training set and test set with 90,000 and 10,000 images respectively. We label each image with the score obtained from the game environment. So the labels are correlated to the number of broken blocks. To train VAE, we use a DCGAN-based architecture, which was originally used to evaluate HWN in Nagano et al. (2019). The detailed architecture is provided in Appendix D.4. To evaluate the models, we measure the correlation between the Poincaré norm of the test images and the labeled scores. The evaluation results and the generated images from the trained models are reported in Table 4 and Figure 5, respectively.

In learning Breakout images, RoWN and the full covariance HWN outperform the diagonal HWN. The isotropic HWN, where the covariance matrix is invariant to any rotation matrix, shows a better correlation than the others in all the settings. However, as reported in Table 14, the test ELBO values are relatively worse than the others. While the isotropic HWN shows a high correlation but relatively lower test ELBO, RoWN shows competitive test ELBO to the others and well aligns the hierarchical structures with respect to the norm in low latent dimensions.

## 5 Related work

**Hyperbolic space for hierarchical representation learning.** Earlier studies on the hierarchical representation learning have focused on modeling explicit hierarchical structures through the Bayesian non-parametrics (Griffiths et al., 2003; Larsen et al., 2002; Salakhutdinov et al., 2012; Teh et al., 2007; Ghahramani et al., 2010; Heller and Ghahramani, 2005) or embedding the hierarchical structure into Euclidean space (Nickel et al., 2011; Grover and Leskovec, 2016; Nguyen et al., 2017). Euclidean

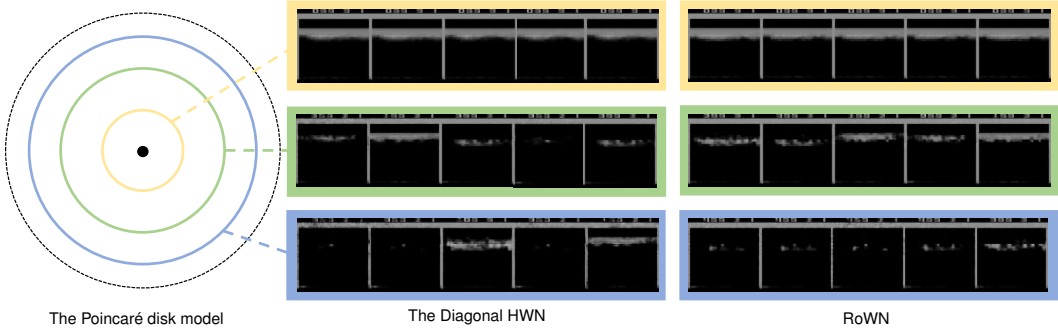

Figure 5: The generated images from a trained VAE endowed with RoWN by using Atari 2600 Breakout images. Every five images are generated from the randomly sampled latent vectors of dimensionality two having the Poincaré norm 0.1, 0.9, 0.95. The larger norm of the sample latent vectors, the more blocks are broken out in the generated images.

space is later shown to require an excessive number of dimensions to embed the original hierarchical structure without any distortion (Linial et al., 1994).

Hyperbolic space has been proposed as an alternative medium to embed the hierarchical data. Theoretically, any tree-structured data can be embedded in hyperbolic space with arbitrarily low distortion (Sarkar, 2011). Based on the theory, learning representation of hierarchical structure in hyperbolic space has been shown great success in various datasets including WordNet hierarchy, graph-structured data, and social network data (Nickel and Kiela, 2017, 2018; Chami et al., 2020; Sun et al., 2020; Zhao et al., 2011; Shavitt and Tankel, 2008). However, these studies are only limited to modeling the explicit hierarchy, which means that the dataset contains an explicit relation between data points. Furthermore, the learning frameworks are limited in the non-probabilistic setting because of the absence of well-behaved distribution in hyperbolic space.

**Distributions in hyperbolic space.** The probabilistic model enables measuring the uncertainty and provides a principled way of learning. To extend the probabilistic learning framework from Euclidean to hyperbolic space or Riemannian manifold in general, one needs to define a well-behaved distribution that has a tractable density and is easy to sample from. Recently, a few studies have proposed probabilistic learning schemes in hyperbolic space (Mathieu et al., 2019; Nagano et al., 2019). Mathieu et al. (2019) introduces parametrizable sampling schemes for the two canonical Gaussian generalizations defined on the Poincaré disk model. The scheme is used to train a hyperbolic VAE and show an improved generalization performance with high interpretability. Nagano et al. (2019) suggests a method of integrating the Bayesian framework with hyperbolic space in the Lorentz model where the simpler closed form of geodesics is defined. Normalizing flow (Rezende and Mohamed, 2015) can be also used to define the hyperbolic distribution in the probabilistic learning framework. Bose et al. (2020) propose two normalizing flows defined on hyperbolic space, which show improvements in learning hierarchical structures in graph data. Mathieu and Nickel (2020) elevate the continuous normalizing flow (Chen et al., 2018) defined on the Euclidean space to arbitrary Riemannian manifold, including the hyperbolic space. Based on Nagano et al. (2019), we analyze the geometric properties of the distribution lying on the Lorentz model and show the limitations of the existing method.

# 6   Limitations

We explore a better method of representing hierarchical data in hyperbolic space. To this end, we propose a simple yet effective alternative of hyperbolic wrapped normal distribution. However, the proposed distribution is limited only to hyperbolic space, and no generalization method for Riemannian space is studied yet. To explore the usefulness of alternative Riemannian spaces, finding a common distribution that can work well in any Riemannian space will be necessary.

RoWN is a subset of the full covariance HWN. However, in many cases, RoWN outperforms the full covariance HWN in our experiments. In general, optimizing the covariance matrix requires learning the quadratic number of parameters with respect to the dimensionality. We conjecture the hardness of optimization leads to the poor performance of the full covariance HWN. To overcome this limitation, a search for a better optimization algorithm in hyperbolic space needs to be explored.

# 7   Conclusions

In this work, we propose a novel method of using RoWN for representing the data with a hierarchical structure. With an in-depth analysis of the geometric properties of HWN, we demonstrate why the common choice of the diagonal covariance matrix for HWN may be inappropriate but the rotated covariance matrix. Our empirical results present that RoWN exhibits better representation ability, both qualitatively and quantitatively, compared to all the baselines: Euclidean normal distribution, diagonal HWN, and full covariance HWN. We hope that our method helps better understanding the anatomy of hyperbolic space and be a promising technique for efficient representation learning.

## Acknowledgement

This work was partly supported by Institute of Information & communications Technology Planning & Evaluation (IITP) grant funded by the Korea government (MSIT) (No.2019-0-01906, Artificial Intelligence Graduate School Program (POSTECH)) and the National Research Foundation of Korea (NRF) grant funded by the Korea government (MSIT) (NRF-2021R1C1C1011375).

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
