# A Proof of Proposition 1

We prove the first proposition in Section 3.1 in this section. Through the proposition, we first show that straight lines that pass through the origin in the Euclidean space are transformed into the geodesics in hyperbolic space by Equation 3.

**Proposition 1.** Suppose $\ell_{\boldsymbol{s}}(t) = t\boldsymbol{s} \in \mathbb{R}^n$ be a line passing through the origin, where $\boldsymbol{s} \in \mathbb{R}^n$ is a directional vector. Then the curve $f_{\boldsymbol{\mu}}(\ell_{\boldsymbol{s}}(t))$ in the Lorentz model $\mathbb{L}^n$ becomes a geodesic.

*Proof.* Let $\boldsymbol{u} := [x_0(\boldsymbol{u}), x_{1:}(\boldsymbol{u})] \in \mathbb{L}^n$ be given, where $x_0 : \mathbb{L}^n \to \mathbb{R}$ and $x_{1:} : \mathbb{L}^n \to \mathbb{R}^n$ denotes the projections, i.e., $x_i(\boldsymbol{u}) = u_i$ . Then,

$$
\begin{aligned}
\mathrm{PT}_{\boldsymbol{0}_{\mathcal{L}} \to \boldsymbol{\mu}}([0, t\boldsymbol{s}]) &= [0, t\boldsymbol{s}] + \frac{1}{x_0(\boldsymbol{\mu}) + 1} \langle \boldsymbol{\mu} - x_0(\boldsymbol{\mu}) \cdot \boldsymbol{0}_{\mathcal{L}}, [0, t\boldsymbol{s}] \rangle_{\mathcal{L}} (\boldsymbol{0}_{\mathcal{L}} + \boldsymbol{\mu}) \\
&= [0, t\boldsymbol{s}] + \frac{1}{x_0(\boldsymbol{\mu}) + 1} \langle [0, x_{1:}(\boldsymbol{\mu})], [0, t\boldsymbol{s}] \rangle_{\mathcal{L}} [x_0(\boldsymbol{\mu}) + 1, x_{1:}(\boldsymbol{\mu})] \\
&= [0, t\boldsymbol{s}] + \frac{1}{x_0(\boldsymbol{\mu}) + 1} \langle x_{1:}(\boldsymbol{\mu}), t\boldsymbol{s} \rangle [x_0(\boldsymbol{\mu}) + 1, x_{1:}(\boldsymbol{\mu})] \\
&= t \left[ \langle x_{1:}(\boldsymbol{\mu}), \boldsymbol{s} \rangle, \boldsymbol{s} + \langle x_{1:}(\boldsymbol{\mu}), \boldsymbol{s} \rangle \frac{x_{1:}(\boldsymbol{\mu})}{x_0(\boldsymbol{\mu}) + 1} \right] .
\end{aligned}
$$

Now, let $\boldsymbol{v} := \left[ \langle x_{1:}(\boldsymbol{\mu}), \boldsymbol{s} \rangle, \boldsymbol{s} + \langle x_{1:}(\boldsymbol{\mu}), \boldsymbol{s} \rangle \frac{x_{1:}(\boldsymbol{\mu})}{x_0(\boldsymbol{\mu})+1} \right]$ and $c := \sqrt{\langle \boldsymbol{v}, \boldsymbol{v} \rangle_{\mathcal{L}}}$. Then,

$$
\begin{aligned}
f_{\boldsymbol{\mu}}(l_{\boldsymbol{s}}) &= \exp_{\boldsymbol{\mu}}(\mathrm{PT}_{\boldsymbol{0}_{\mathcal{L}} \to \boldsymbol{\mu}}((0, t\boldsymbol{s}))) \\
&= \exp_{\boldsymbol{\mu}}(t\boldsymbol{v}) \\
&= \cosh(ct)\boldsymbol{\mu} + \sinh(ct)\frac{\boldsymbol{v}}{c}.
\end{aligned}
$$

Recall that the geodesic of the Lorentz model is $\cosh(t)\boldsymbol{x} + \sinh(t)\boldsymbol{y}$ where $\boldsymbol{x} \in \mathbb{L}^n$, $\langle \boldsymbol{y}, \boldsymbol{y} \rangle_{\mathcal{L}} = 1$ and $\boldsymbol{y} \in \mathcal{T}_{\boldsymbol{x}}\mathbb{L}^n$ (Robbin and Salamon, 2022). $\square$

The proposition indicates that every straight line that passes through the origin including the principal axes, is transformed into a geodesic in hyperbolic space.

# B Proof of Proposition 2

We prove the second proposition in Section 3.1 to show the geometrical properties of HWN. Based on the first proposition, we provide our main proposition, which fully characterizes the structure of principal axes when projected to the Poincaré disk model:

**Proposition 2.** Let $\mathrm{Proj}(\boldsymbol{u})$ be the projection function from Lorentz model to Poincaré model, i.e., $\mathrm{Proj}(\boldsymbol{u}) = \frac{x_{1:}(\boldsymbol{u})}{x_0(\boldsymbol{u})+1}, \forall \boldsymbol{u} \in \mathbb{L}^n$. Let $\ell_{\boldsymbol{s}}$ be a principal axis of the normal distribution defined in $\mathbb{R}^n$ and $\boldsymbol{\mu}$ the mean of HWN in $\mathbb{L}^n$, then $\boldsymbol{s}$ is the tangent vector of $\mathrm{Proj}(f_{\boldsymbol{\mu}}(\ell_{\boldsymbol{s}}))$ on $\mathrm{Proj}(\boldsymbol{\mu})$.

*Proof.* First, we project the transformed principal axis to the Poincaré disk model as:

$$
\begin{aligned}
\mathrm{Proj}\left(f_{\boldsymbol{\mu}}(l_{\boldsymbol{s}})\right) &= \mathrm{Proj}\left(\cosh(ct)\boldsymbol{\mu} + \sinh(ct)\frac{\boldsymbol{v}}{c}\right) \\
&= \mathrm{Proj}\left(\left[\cosh(ct)x_0(\boldsymbol{\mu}) + \sinh(ct)\frac{x_0(\boldsymbol{v})}{c}, \cosh(ct)x_{1:}(\boldsymbol{\mu}) + \sinh(ct)\frac{x_{1:}(\boldsymbol{v})}{c}\right]\right) \\
&= \frac{\cosh(ct)x_{1:}(\boldsymbol{\mu}) + \sinh(ct)x_{1:}(\boldsymbol{v})/c}{\cosh(ct)x_0(\boldsymbol{\mu}) + \sinh(ct)x_0(\boldsymbol{v})/c + 1}.
\end{aligned}
$$

Then, the derivative of the projected curve with respect to $t$ is derived as:

$$
\begin{aligned}
\frac{\partial}{\partial t}\mathrm{Proj}\left(f_{\boldsymbol{\mu}}(l_{\boldsymbol{s}})\right) &= \frac{\partial}{\partial t}\frac{\cosh(ct)x_{1:}(\boldsymbol{\mu}) + \sinh(ct)x_{1:}(\boldsymbol{v})/c}{\cosh(ct)x_0(\boldsymbol{\mu}) + \sinh(ct)x_0(\boldsymbol{v})/c + 1} \\
&= \frac{\sinh(ct)x_{1:}(\boldsymbol{\mu})c + \cosh(ct)x_{1:}(\boldsymbol{v})}{\cosh(ct)x_0(\boldsymbol{\mu}) + \sinh(ct)x_0(\boldsymbol{v})/c + 1} \\
&\quad - \frac{(\sinh(ct)x_0(\boldsymbol{\mu})c + \cosh(ct)x_0(\boldsymbol{v}))(\cosh(ct)x_{1:}(\boldsymbol{\mu}) + \sinh(ct)x_{1:}(\boldsymbol{v})/c)}{(\cosh(ct)x_0(\boldsymbol{\mu}) + \sinh(ct)x_0(\boldsymbol{v})/c + 1)^2}.
\end{aligned}
$$

As $\mathrm{Proj}(\boldsymbol{\mu})$ is the point of the curve at $t = 0$, the tangent vector of the curve on $\mathrm{Proj}(\boldsymbol{\mu})$ can be computed by substituting $t = 0$:

$$
\begin{aligned}
\frac{\partial}{\partial t}\mathrm{Proj}\left(f_{\boldsymbol{\mu}}(l_{\boldsymbol{s}})\right)\Big|_{t=0} &= \frac{x_{1:}(\boldsymbol{v})}{x_0(\boldsymbol{\mu}) + 1} - \frac{x_0(\boldsymbol{v})x_{1:}(\boldsymbol{\mu})}{(x_0(\boldsymbol{\mu}) + 1)^2} \\
&= \frac{\boldsymbol{s}}{x_0(\boldsymbol{\mu}) + 1} + \frac{\langle x_{1:}(\boldsymbol{\mu}), \boldsymbol{s}\rangle x_{1:}(\boldsymbol{\mu})}{(x_0(\boldsymbol{\mu}) + 1)^2} - \frac{\langle x_{1:}(\boldsymbol{\mu}), \boldsymbol{s}\rangle x_{1:}(\boldsymbol{\mu})}{(x_0(\boldsymbol{\mu}) + 1)^2} \\
&= \frac{\boldsymbol{s}}{x_0(\boldsymbol{\mu}) + 1} \\
&\propto \boldsymbol{s}.
\end{aligned}
$$

$\square$

The proposition reveals that the principal axes of the HWN are locally parallel to the standard bases in the Poincaré disk model. To visualize the proposition, we plot the contour line and the principal axes of the two-dimensional diagonal covariance normal distribution before and after the transformation in Figure 2. We observe that the tangent lines of the transformed principal axes are parallel to the standard bases in hyperbolic space.

# C  Details of Rotated Hyperbolic Wrapped Normal Distribution

The details of RoWN, i.e. sampling and the probability density computation, are described in this section. We start with a mean vector $\boldsymbol{\mu} \in \mathbb{L}^n$ and a diagonal covariance matrix $\Sigma$ as in the standard HWN. Based on the mean vector, we construct RoWN by rotating the covariance matrix as follows:

1. Compute the rotation matrix $\boldsymbol{R}$ that rotates the x-axis ($[\pm 1, \ldots, 0] \in \mathbb{R}^n$) to $\boldsymbol{\mu}_{1:}$.
2. Substitute the covariance matrix of Gaussian normal with $\boldsymbol{R}\Sigma\boldsymbol{R}^T$.

Thus, the rotation matrix $\boldsymbol{R}$, which rotates a unit vector from $\boldsymbol{x}$ to $\boldsymbol{y}$, can be computed as:

$$\boldsymbol{R} = \boldsymbol{I} + (\boldsymbol{y}^T\boldsymbol{x} - \boldsymbol{x}^T\boldsymbol{y}) + \frac{1}{1 + \langle \boldsymbol{x}, \boldsymbol{y} \rangle}(\boldsymbol{y}^T\boldsymbol{x} - \boldsymbol{x}^T\boldsymbol{y})^2. \tag{6}$$

Algorithm 2 shows the entire algorithm of constructing RoWN.

Note that the construction is straightforward but still keeps the following benefits of the HWN: 1) The sampling can be done efficiently, and 2) the computation of the probability density of the samples is tractable. As the HWN provides a tractable probability density function for any kind of covariance matrix, we can easily compute the probability density of a given sample from RoWN. For the details of sampling and probability density computation, see Algorithm 3 & 4.

---

**Algorithm 2** RoWN($\boldsymbol{\mu}, \Sigma$)

---

**Input** Mean $\boldsymbol{\mu} \in \mathbb{L}^n$, diagonal covariance matrix $\Sigma \in \mathbb{R}^{n \times n}$

**Output** Rotated covariance matrix $\hat{\Sigma}$.

1: $\boldsymbol{x} = [\pm 1, \ldots, 0] \in \mathbb{R}^n, \boldsymbol{y} = \boldsymbol{\mu}_{1:}/\|\boldsymbol{\mu}_{1:}\|$               $\triangleright \pm$ is determined by the sign of $\boldsymbol{\mu}_0$
2: $\boldsymbol{R} = \boldsymbol{I} + (\boldsymbol{y}^T\boldsymbol{x} - \boldsymbol{x}^T\boldsymbol{y}) + (\boldsymbol{y}^T\boldsymbol{x} - \boldsymbol{x}^T\boldsymbol{y})^2/(1 + \langle \boldsymbol{x}, \boldsymbol{y} \rangle)$
3: **return** $\hat{\Sigma} = \boldsymbol{R}\Sigma\boldsymbol{R}^T$

---

---

**Algorithm 3** Sampling process with the rotated hyperbolic wrapped normal distribution

---

**Input** Mean $\boldsymbol{\mu} \in \mathbb{L}^n$, diagonal covariance matrix $\Sigma \in \mathbb{R}^{n \times n}$

**Output** Sample $\boldsymbol{z} \in \mathbb{L}^n$

1: Construct $\hat{\Sigma} = \text{RoWN}(\boldsymbol{\mu}, \Sigma)$
2: Sample $\boldsymbol{v} \sim \mathcal{N}(\boldsymbol{0}, \hat{\Sigma})$
3: **return** $\boldsymbol{z} = f_{\boldsymbol{\mu}}(\boldsymbol{v})$

---

---

**Algorithm 4** Probability density computation of the rotated hyperbolic wrapped normal distribution

---

**Input** Mean $\boldsymbol{\mu} \in \mathbb{L}^n$, diagonal covariance matrix $\Sigma \in \mathbb{R}^{n \times n}$, sample $\boldsymbol{z} \in \mathbb{L}^n$

**Output** Log probability of $\boldsymbol{z}$.

1: Construct $\hat{\Sigma} = \text{RoWN}(\boldsymbol{\mu}, \Sigma)$
2: $\boldsymbol{u} = \log_{\boldsymbol{\mu}}(\boldsymbol{z})$
3: $\boldsymbol{v} = \text{PT}_{\boldsymbol{\mu} \to \boldsymbol{0}_{\mathcal{L}}}(\boldsymbol{u})$                                        $\triangleright \boldsymbol{v} = f_{\boldsymbol{\mu}}^{-1}(\boldsymbol{z})$
4: **return** (log probability of $\boldsymbol{v}_{1:}$ from $\mathcal{N}(\boldsymbol{0}, \hat{\Sigma})$) $- (n-1)(\log \sinh \|\boldsymbol{u}\|_{\mathcal{L}} - \log \|\boldsymbol{u}\|_{\mathcal{L}})$

---

# D Experimental Details

In this section, we provide the details of the experiments in Section 4.

## D.1 Baselines

For all the experiments, we compare the performance of RoWN with four baselines: the normal distribution in the Euclidean space, the isotropic HWN (Nagano et al., 2019), the diagonal HWN, and the full covariance HWN.

In the process of constructing the distributions for each application, i.e. the variational distribution of VAE and the embedding distribution of probabilistic word embedding, the distributions except the full covariance HWN have a diagonal matrix as an input, and then the softplus operation is used to make it positive. The full covariance HWN has a 2D matrix $\Sigma \in \mathbb{R}^{n \times n}$ as an input and constructs a covariance matrix as $\Sigma\Sigma^T + \epsilon \boldsymbol{I}$, to match the positive-definite property, where $\epsilon$ is set to $1\mathrm{e}{-}9$ in our experiments. For the mean value, we concatenate zero at the first dimension and transport it to the Lorentz model using $\exp_{\mathbf{0}_{\mathcal{L}}}$.

For the hyperbolic VAE models, we use $\log_{\mathbf{0}_{\mathcal{L}}}$ to transform the input of the decoder to the Euclidean space, as suggested in Mathieu et al. (2019).

## D.2 Noisy synthetic binary tree

**Experimental setting.** A synthetic binary tree dataset is first used to show the performance of representing hierarchy in Nagano et al. (2019), where each node in a tree corresponds to a sequence of binary values. Figure 4a shows an example of the depth three binary tree, where a parent and child only differ in one digit. We add spherical noises to the nodes in the same level of hierarchy as described in Figure 4a as the noisy samples. With the data points with additional noises, we can create a dataset containing a local-level variation in the hierarchy. For the experiments, we uniformly sample the spherical noise from $[0, \pi/4]$.

We train hyperbolic VAE on *noisy synthetic binary tree* with varying depth from 4 to 7. For each depth $d$, we use a three-layer fully connected neural network as the architecture where the number of hidden units is $2^{d+3}$ and the latent dimension is $d$. We use ReLU as the activation function for each layer except the last layer of the encoder and decoder. The overall architecture is shown in Table 5 and Table 6. We use a Gaussian negative log-likelihood loss for the reconstruction loss with fixed $\sigma = 0.01$, which is selected for sufficient reconstruction performance on the train set.

Table 5: Encoder architecture for *noisy synthetic binary tree*

| Layer | Output dim | Activation |
|-------|-----------|------------|
| FC    | $2^{d+3}$ | ReLU       |
| FC    | $2^{d+3}$ | ReLU       |
| FC    | $2d$      | None       |

Table 6: Decoder architecture for *noisy synthetic binary tree*

| Layer | Output dim  | Activation |
|-------|-------------|------------|
| FC    | $2^{d+3}$   | ReLU       |
| FC    | $2^{d+3}$   | ReLU       |
| FC    | $2^d - 1$   | None       |

**Results.** We report 1) the correlation between the hamming distance and the embedding distance and 2) the correlation between the depth and the norm of the embeddings. The first correlation is computed over all possible pairs of test points. For the norm of the hyperbolic embeddings, we use the Poincaré norm, which can be calculated by projecting the Lorentz model embedding to the Poincaré disk model.

As Table 7 shows, while all the models show similar performance with respect to the ELBO, the full covariance HWN and RoWN show better performance than the diagonal HWN except depth six, outperforming the Euclidean model in every setting. RoWN preserves the depth information better than the other distributions in general. We additionally visualize the variational mean obtained by training the tree of depth three in Figure 4b, where the hierarchical structure is well preserved in the hyperbolic embedding space.

Table 7: Results of *noisy synthetic binary tree*. The results are averaged over 10 runs. The hyperbolic models outperform the Euclidean model in all settings. Overall, RoWN preserves the hierarchical information better than the other distributions.

| | | depth | | | |
|---|---|---|---|---|---|
| | | 4 | 5 | 6 | 7 |
| Correlation w/ distance | Euclidean | $0.748_{\pm.032}$ | $0.740_{\pm.013}$ | $0.741_{\pm.008}$ | $0.733_{\pm.014}$ |
| | HWN (isotropic $\Sigma$) | $0.773_{\pm.030}$ | $0.809_{\pm.016}$ | $0.798_{\pm.008}$ | $0.735_{\pm.022}$ |
| | HWN (diagonal $\Sigma$) | $0.814_{\pm.008}$ | $0.791_{\pm.023}$ | $0.817_{\pm.010}$ | $0.759_{\pm.025}$ |
| | HWN (full $\Sigma$) | $0.827_{\pm.015}$ | $0.798_{\pm.026}$ | $0.798_{\pm.010}$ | $0.794_{\pm.014}$ |
| | RoWN | $0.820_{\pm.015}$ | $0.807_{\pm.017}$ | $0.822_{\pm.017}$ | $0.788_{\pm.016}$ |
| Correlation w/ depth | Euclidean | $0.762_{\pm.117}$ | $0.807_{\pm.038}$ | $0.712_{\pm.054}$ | $0.612_{\pm.049}$ |
| | HWN (isotropic $\Sigma$) | $0.902_{\pm.033}$ | $0.867_{\pm.034}$ | $0.811_{\pm.029}$ | $0.602_{\pm.066}$ |
| | HWN (diagonal $\Sigma$) | $0.918_{\pm.028}$ | $0.808_{\pm.076}$ | $0.862_{\pm.035}$ | $0.697_{\pm.076}$ |
| | HWN (full $\Sigma$) | $0.956_{\pm.015}$ | $0.878_{\pm.051}$ | $0.870_{\pm.033}$ | $0.815_{\pm.055}$ |
| | RoWN | $0.930_{\pm.026}$ | $0.911_{\pm.027}$ | $0.901_{\pm.034}$ | $0.827_{\pm.047}$ |
| Test ELBO | Euclidean | $22.591_{\pm.183}$ | $55.168_{\pm.092}$ | $124.374_{\pm.093}$ | $266.854_{\pm.199}$ |
| | HWN (isotropic $\Sigma$) | $22.026_{\pm.201}$ | $54.054_{\pm.158}$ | $122.981_{\pm.145}$ | $265.316_{\pm.217}$ |
| | HWN (diagonal $\Sigma$) | $22.480_{\pm.144}$ | $54.540_{\pm.117}$ | $123.444_{\pm.110}$ | $265.704_{\pm.154}$ |
| | HWN (full $\Sigma$) | $22.371_{\pm.136}$ | $55.032_{\pm.141}$ | $124.125_{\pm.189}$ | $266.499_{\pm.112}$ |
| | RoWN | $22.354_{\pm.138}$ | $54.648_{\pm.142}$ | $123.606_{\pm.066}$ | $266.146_{\pm.112}$ |

**Decomposition of the radial and angular dependency.** To show how well the angular and radial representations are decomposed, we compute the Pearson correlation between the radial axis and the angular axis, which has been shown to be an effective measure of variable dependency in disentangled representation learning (Jo and Seo, 2019; Horan et al., 2021). Table 8 shows that the absolute value of the correlation is lower or similar to 0.1 in all the models, including RoWN.

**Learnable rotation.** We add experiments for the models that learn the rotation direction, which is originally fixed to the direction of $\boldsymbol{\mu}$ (in Algorithm 1, the $\boldsymbol{y}$ vector). Given data, the encoder gives the rotation direction. We test the models on the noisy synthetic binary tree setting in our paper. As shown Table 9, as the depth becomes deeper, the learnable rotation models usually underperform RoWN. Figure 6 shows behaviors of learned representation by an alternative Learnable Rotation 1. Most of the rotation directions are pointing or orthogonal (black line segments on each node) to the direction of its parent node. This implies that the alternatives of RoWN can learn representations of nodes to align not to the root node but to the parent node. We note that these alternatives work well with shallow depths but not with great depths.

### D.3 WordNet

**Experimental setting.** We train a probabilistic word embedding model with WordNet dataset (Fellbaum, 1998). We initialize the mean and variance parameters with $\mathcal{N}(\mathbf{0}, 0.01)$. For the full covariance model, we use a learning rate 0.01. For the other models, we set the learning rate 0.015 for the first 100 epochs and then set the learning rate to 0.6 for the remaining steps as done in (Nickel and Kiela, 2017; Nagano et al., 2019). We evaluate the learned representations by computing the average rank of all the hypernymies. The rank of a given pair of words $s$ and $t$ is computed among the distances between all possible pairs of the words $s$ and $t'$ without hypernymy.

**Results.** We evaluate the learned representations by computing the average rank of all the hypernymies. The rank of a given pair of words $s$ and $t$ is computed among the distances between all possible pairs of the words $s$ and $t'$ without hypernymy. Table 2 shows the empirical performances of representing the word data. We report the performance with the mean rank (MR) and the mean average precision (mAP). RoWN preserves the hierarchical structure better than the other distributions, while the full covariance HWN fails due to unstable optimization.

Table 8: Correlation between the radian axis and the angular axis.

| | | depth | | | |
|---|---|---|---|---|---|
| | | 4 | 5 | 6 | 7 |
| Correlation btw. $r$ and $\theta_1$ | Euclidean | $0.144_{\pm.170}$ | $0.007_{\pm.105}$ | $0.039_{\pm.106}$ | $-0.026_{\pm.093}$ |
| | HWN (diagonal $\Sigma$) | $0.025_{\pm.150}$ | $0.015_{\pm.137}$ | $0.110_{\pm.065}$ | $-0.003_{\pm.134}$ |
| | HWN (full $\Sigma$) | $-0.053_{\pm.216}$ | $0.049_{\pm.136}$ | $-0.017_{\pm.169}$ | $0.012_{\pm.066}$ |
| | RoWN | $0.080_{\pm.163}$ | $0.030_{\pm.097}$ | $0.112_{\pm.093}$ | $0.025_{\pm.083}$ |
| Correlation btw. $r$ and $\theta_2$ | Euclidean | $0.116_{\pm.232}$ | $-0.039_{\pm.210}$ | $0.109_{\pm.131}$ | $0.006_{\pm.095}$ |
| | HWN (diagonal $\Sigma$) | $0.066_{\pm.170}$ | $-0.021_{\pm.190}$ | $0.044_{\pm.122}$ | $-0.024_{\pm.115}$ |
| | HWN (full $\Sigma$) | $0.297_{\pm.116}$ | $0.113_{\pm.109}$ | $0.013_{\pm.122}$ | $0.061_{\pm.103}$ |
| | RoWN | $0.013_{\pm.178}$ | $0.025_{\pm.173}$ | $-0.004_{\pm.115}$ | $0.064_{\pm.082}$ |
| Correlation btw. $r$ and $\theta_3$ | Euclidean | $0.067_{\pm.220}$ | $-0.110_{\pm.220}$ | $-0.016_{\pm.159}$ | $0.011_{\pm.098}$ |
| | HWN (diagonal $\Sigma$) | $0.123_{\pm.252}$ | $-0.139_{\pm.123}$ | $-0.019_{\pm.133}$ | $-0.095_{\pm.101}$ |
| | HWN (full $\Sigma$) | $-0.053_{\pm.144}$ | $-0.088_{\pm.107}$ | $0.106_{\pm.120}$ | $0.024_{\pm.079}$ |
| | RoWN | $0.127_{\pm.253}$ | $-0.120_{\pm.169}$ | $-0.012_{\pm.120}$ | $-0.012_{\pm.083}$ |
| Correlation btw. $r$ and $\theta_4$ | Euclidean | - | $-0.015_{\pm.073}$ | $0.013_{\pm.081}$ | $0.026_{\pm.065}$ |
| | HWN (diagonal $\Sigma$) | - | $-0.047_{\pm.117}$ | $-0.035_{\pm.147}$ | $0.080_{\pm.096}$ |
| | HWN (full $\Sigma$) | - | $0.079_{\pm.150}$ | $0.042_{\pm.108}$ | $0.086_{\pm.102}$ |
| | RoWN | - | $-0.031_{\pm.116}$ | $-0.070_{\pm.112}$ | $0.086_{\pm.115}$ |
| Correlation btw. $r$ and $\theta_5$ | Euclidean | - | - | $0.082_{\pm.075}$ | $-0.029_{\pm.109}$ |
| | HWN (diagonal $\Sigma$) | - | - | $0.022_{\pm.101}$ | $-0.041_{\pm.097}$ |
| | HWN (full $\Sigma$) | - | - | $-0.058_{\pm.111}$ | $0.076_{\pm.064}$ |
| | RoWN | - | - | $-0.016_{\pm.111}$ | $-0.030_{\pm.112}$ |
| Correlation btw. $r$ and $\theta_6$ | Euclidean | - | - | - | $-0.018_{\pm.073}$ |
| | HWN (diagonal $\Sigma$) | - | - | - | $-0.030_{\pm.082}$ |
| | HWN (full $\Sigma$) | - | - | - | $0.035_{\pm.103}$ |
| | RoWN | - | - | - | $0.006_{\pm.067}$ |

**Optimization issue in training full covariance HWN.** In the results, we find that the full covariance HWN shows poor performance compared to the other models. We run extensive experiments to show that the full covariance HWN is difficult to optimize especially in WordNet. Figure 7 shows the performance of the full covariance HWN with varying hyperparameters, i.e. learning rate, the burn-in factor, and the initialization method, which seems to be poor whatever we choose. We conducted

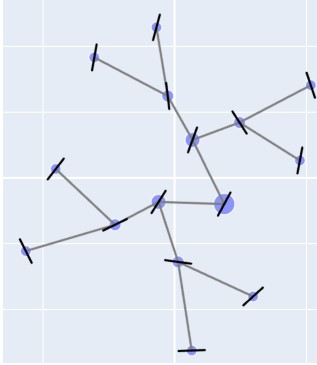

Figure 6: Visualization of the representations from learnable rotation model. The representations are from the Learnable Rotation 1 model learned on the depth 4 noisy synthetic binary tree with latent dimension 2. The size of the circles denotes the depth, where the biggest circle denotes the root node. Most of the rotation directions are pointing or orthogonal (black line segments on each node) to the direction of its parent node.

Table 9: Results of learnable rotation models. Learnable Rotation 1 model outputs the rotation direction parallel to the mean and variance, while Learnable Rotation 2 model outputs the rotation direction by feeding the mean to a fully connected layer.

| | | depth | | | |
|---|---|---|---|---|---|
| | | 4 | 5 | 6 | 7 |
| Correlation w/ distance | RoWN | $0.820_{\pm.015}$ | $0.807_{\pm.017}$ | $0.822_{\pm.017}$ | $0.788_{\pm.016}$ |
| | Learnable Rotation 1 | $0.820_{\pm.013}$ | $0.804_{\pm.016}$ | $0.808_{\pm.013}$ | $0.770_{\pm.028}$ |
| | Learnable Rotation 2 | $0.817_{\pm.018}$ | $0.811_{\pm.018}$ | $0.801_{\pm.008}$ | $0.772_{\pm.021}$ |
| Correlation w/ depth | RoWN | $0.930_{\pm.026}$ | $0.911_{\pm.027}$ | $0.901_{\pm.034}$ | $0.827_{\pm.047}$ |
| | Learnable Rotation 1 | $0.952_{\pm.018}$ | $0.903_{\pm.037}$ | $0.845_{\pm.057}$ | $0.711_{\pm.072}$ |
| | Learnable Rotation 2 | $0.948_{\pm.015}$ | $0.907_{\pm.019}$ | $0.844_{\pm.013}$ | $0.724_{\pm.058}$ |
| Test ELBO | RoWN | $22.354_{\pm.138}$ | $54.648_{\pm.142}$ | $123.606_{\pm.066}$ | $266.146_{\pm.112}$ |
| | Learnable Rotation 1 | $22.342_{\pm.097}$ | $54.741_{\pm.111}$ | $123.619_{\pm.180}$ | $266.076_{\pm.159}$ |
| | Learnable Rotation 2 | $22.286_{\pm.120}$ | $54.529_{\pm.067}$ | $123.382_{\pm.161}$ | $265.962_{\pm.217}$ |

an additional analysis on what causes the optimization instability and found that the number of samples used to approximate the KL divergence is critical to full covariance HWN, especially in the Wordnet dataset. In VAE, we can observe more stable results. We speculate that the stability improved since the relatively simple prior (standard normal distribution) is employed with the full covariance variational distribution. Table 11 shows that as the number of training samples increases, the performance increases, but the result is still poor, and using more training samples leads to an additional computation time.

## D.4 Atari 2600 Breakout

**Experimental setting.** To learn the implicit hierarchy that can be observed from the trajectories of Breakout, we train the VAE models with the Atari 2600 Breakout images. The images of Breakout are collected by using a pre-trained Deep Q-network (Mnih et al., 2015) and divided into a training set and test set with 90,000 and 10,000 images respectively. We label each image with the score obtained from the game environment. So the labels are correlated to the number of broken blocks. To train VAE, we use a DCGAN-based architecture, which was originally used to evaluate HWN in Nagano et al. (2019). The detailed architecture is provided in Table 12 and Table 13. We use binary cross-entropy loss for the reconstruction loss.

Table 10: Results of *WordNet*. The results are an average of 5 runs. Based on the rank of hypernymy pairs among non-hypernymy pairs, we report the mean rank (MR) and mean average precision (mAP) for evaluation.

| | | latent dimension | | |
|---|---|---|---|---|
| | | 5 | 10 | 20 |
| MR | Euclidean | $13.968_{\pm0.504}$ | $3.862_{\pm0.281}$ | $1.955_{\pm0.157}$ |
| | HWN (isotropic $\Sigma$) | $14.568_{\pm2.203}$ | $4.470_{\pm0.669}$ | $3.125_{\pm0.455}$ |
| | HWN (diagonal $\Sigma$) | $16.590_{\pm1.146}$ | $3.891_{\pm0.447}$ | $2.062_{\pm0.088}$ |
| | HWN (full $\Sigma$) | $557.309_{\pm18.006}$ | $466.513_{\pm75.142}$ | $599.140_{\pm18.916}$ |
| | RoWN | $16.271_{\pm2.985}$ | $2.888_{\pm0.162}$ | $1.783_{\pm0.090}$ |
| mAP | Euclidean | $0.565_{\pm0.014}$ | $0.801_{\pm0.020}$ | $0.902_{\pm0.008}$ |
| | HWN (isotropic $\Sigma$) | $0.617_{\pm0.012}$ | $0.820_{\pm0.013}$ | $0.847_{\pm0.017}$ |
| | HWN (diagonal $\Sigma$) | $0.565_{\pm0.020}$ | $0.805_{\pm0.015}$ | $0.905_{\pm0.007}$ |
| | HWN (full $\Sigma$) | $0.032_{\pm0.003}$ | $0.063_{\pm0.005}$ | $0.079_{\pm0.021}$ |
| | RoWN | $0.593_{\pm0.024}$ | $0.844_{\pm0.009}$ | $0.921_{\pm0.005}$ |

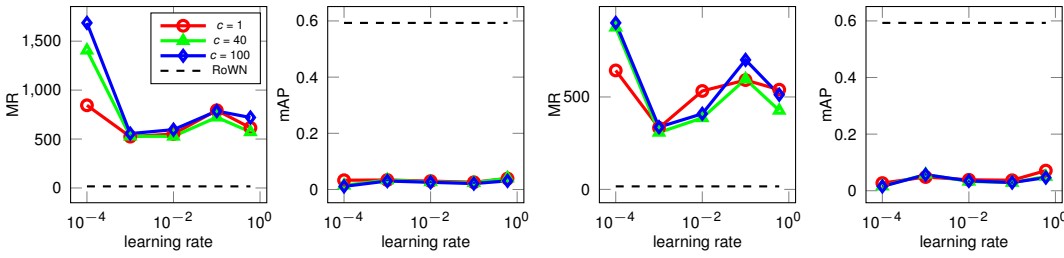

(a) MR of full initializa-
tion

(b) mAP of full initializa-
tion

(c) MR of diagonal initial-
ization

(d) mAP of diagonal ini-
tialization

Figure 7: Varying hyperparameters of the full covariance HWN on WordNet. We run several hyper-parameters combination of full covariance HWN on WordNet. We fix the latent dimension to 5 and burn-in epochs to 100. We vary the learning rate from $1e{-}4$ to 0.6 and the factor $c$, which is used to reduce the learning rate in the burn-in steps, from 1 to 100. We run 10,000 epochs. (a,b) When we initialize the entire covariance matrix with $\mathcal{N}(0, 0.01)$, the full covariance HWN show poor performance with any hyper-parameter combinations. (c,d) Initializing only the diagonal entries with $\mathcal{N}(0, 0.01)$ and let the remaining part to zero improves the performance of the full covariance HWN but it still performs worse than RoWN.

Table 11: Varying samples of the full covariance HWN on WordNet.

| # of samples | 1 | 50 | 100 |
|---|---|---|---|
| MR | $326.185_{\pm 7.610}$ | $73.572_{\pm 9.926}$ | $69.964_{\pm 10.457}$ |
| mAP | $0.056_{\pm 0.004}$ | $0.241_{\pm 0.021}$ | $0.241_{\pm 0.019}$ |
| Runtime (s/epoch) | 6.288 | 6.748 | 8.163 |

**Results.** To evaluate the models, we measure the correlation between the norm of the test images and the labeled scores. For the norm of the hyperbolic embeddings, we use the Poincaré norm, which can be calculated by projecting the Lorentz model embedding to the Poincaré disk model. The results are reported in Table 14. While all the models show similar representation power with respect to ELBO, RoWN and the full covariance HWN outperform the diagonal HWN. Especially, RoWN aligns the hierarchical structures better with respect to the norm in low latent dimensions.

Nagano et al. (2019) report a higher score of correlation in latent dimension 20, but we find some issues with the result. First, Nagano et al. (2019) compute the correlation between the labeled scores and the norm in the tangent space ($v$ vector from Algorithm 4), not the Poincaré norm. The projection function in Proposition 2 depends on the first element of the input vector. Thus the Poincaré norm is not proportional to the $v$ norm, and computing the correlation with the $v$ norm will show different behavior compared to the correlation with the Poincaré norm. Second, the reproduction results obtained from the code by the official repository[1] are far from the reported correlation. Our reproduction results of the correlation between the labeled scores and the $v$ norm show 0.616, and between the Poincaré norm show 0.501 averaged over four runs.

**Qualitative results.** Figure 8 shows more examples of generated images from VAE models trained on Breakout images with two dimensional latent space.

---

[1] https://github.com/pfnet-research/hyperbolic_wrapped_distribution

Table 12: Encoder architecture for Breakout

| Layer | Output dim | Activation |
|---|---|---|
| Conv2d | $80 \times 80 \times 16$ | ReLU |
| Conv2d | $40 \times 40 \times 32$ | ReLU |
| Conv2d | $40 \times 40 \times 32$ | ReLU |
| Conv2d | $20 \times 20 \times 64$ | ReLU |
| Conv2d | $20 \times 20 \times 64$ | ReLU |
| Conv2d | $10 \times 10 \times 64$ | ReLU |
| FC | $2 \times$ latent dimension | None |

Table 13: Decoder architecture for Breakout

| Layer | Output dim | Activation |
|---|---|---|
| FC | $10 \times 10 \times 64$ | ReLU |
| ConvTranspose2d | $20 \times 20 \times 32$ | ReLU |
| Conv2d | $20 \times 20 \times 32$ | ReLU |
| ConvTranspose2d | $40 \times 40 \times 16$ | ReLU |
| Conv2d | $40 \times 40 \times 16$ | ReLU |
| ConvTranspose2d | $80 \times 80 \times 1$ | Sigmoid |

Table 14: Results of *Atari 2600 Breakout*. The results are averaged over 10 runs. We measure the correlation between the score of an image and the Poincaré norm of the variational mean.

| | | latent dimension | | |
|---|---|---|---|---|
| | | 10 | 15 | 20 |
| Correlation btw. score and norm | Euclidean | $0.379_{\pm.007}$ | $0.436_{\pm.029}$ | $0.479_{\pm.020}$ |
| | HWN (isotropic $\Sigma$) | $0.513_{\pm.012}$ | $0.598_{\pm.021}$ | $0.607_{\pm.015}$ |
| | HWN (diagonal $\Sigma$) | $0.478_{\pm.011}$ | $0.513_{\pm.006}$ | $0.513_{\pm.008}$ |
| | HWN (full $\Sigma$) | $0.483_{\pm.011}$ | $0.520_{\pm.009}$ | $0.563_{\pm.010}$ |
| | RoWN | $0.497_{\pm.014}$ | $0.556_{\pm.014}$ | $0.561_{\pm.029}$ |
| Test ELBO | Euclidean | $-1269.044_{\pm.241}$ | $-1269.624_{\pm.258}$ | $-1269.682_{\pm.178}$ |
| | HWN (isotropic $\Sigma$) | $-1271.018_{\pm.440}$ | $-1272.139_{\pm.170}$ | $-1272.914_{\pm.118}$ |
| | HWN (diagonal $\Sigma$) | $-1269.816_{\pm.272}$ | $-1270.725_{\pm.260}$ | $-1271.087_{\pm.234}$ |
| | HWN (full $\Sigma$) | $-1269.021_{\pm.320}$ | $-1269.569_{\pm.206}$ | $-1269.882_{\pm.438}$ |
| | RoWN | $-1269.531_{\pm.212}$ | $-1270.203_{\pm.211}$ | $-1270.967_{\pm.183}$ |

# E   Discussion on Root Placement

Note that due to the isometry of the geometry, the root node can be placed anywhere in hyperbolic space. In other words, infinitely many sets of embeddings preserve the same pairwise distances between the nodes. This reveals that finding the appropriate isometry, where the root node is placed near the origin, is important for using RoWN as the distribution. In this section, we discuss the techniques we used to place the root node near the origin of each application.

## E.1   Probabilistic word embedding model

In our experiments, to place the root node near the origin, we have initialized embeddings from $\mathcal{N}(0, 0.01I)$, which are then moved to the Lorentz model using the exponential map, with learning rate warm-up Nagano et al. (2019). The results with different initialization method are shown in Table 3.

## E.2   Hyperbolic VAE

When RoWN is used as a variational distribution in the hyperbolic VAE, the application of RoWN is different from the probabilistic word embedding model since the KL divergence encourages all variational means to be close to the prior mean. Suppose we only focus on nodes at a certain depth in a tree. In that case, it can be easily identified that it would be beneficial to have all nodes at the same level of the norm to minimize the geometric mean between the prior and posterior means. Here, we assume that each pair of nodes requires to have a certain amount of distance to minimize the reconstruction error. As shown in Figure 3a, the original HWN is difficult to have the nodes at the same depth with similar norms since the local variation cannot be modeled through the radial direction. Eventually, the root node slightly deviates from the prior mean. With RoWN, as shown in

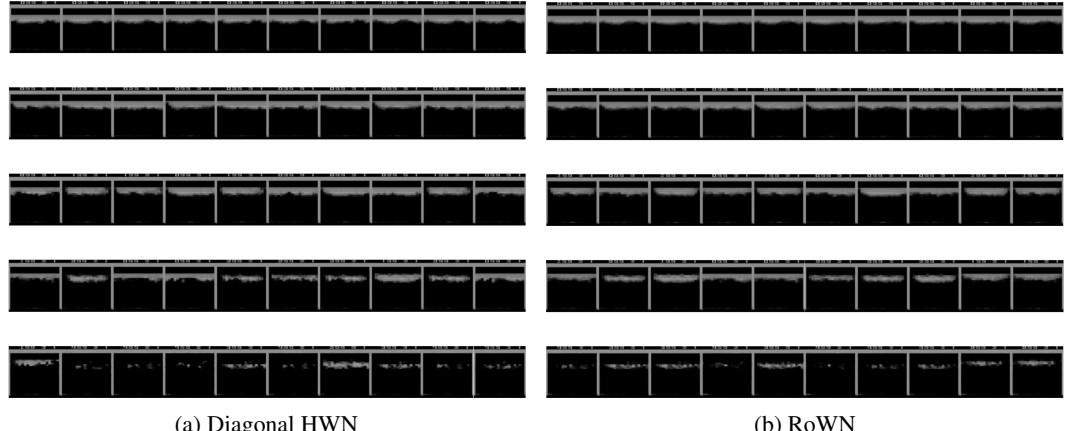

| (a) Diagonal HWN | (b) RoWN |

Figure 8: Generation results of the VAE models trained on Breakout images. The models are trained on Breakout images with two dimensional latent space. We generate the images from randomly sampled latent vectors having Poincaré norm as 0.1, 0.3, 0.5, 0.7, 0.9.

Figure 3c, all the nodes at the same depth can be placed at a similar norm while preserving their local variations. If this is indeed the case, the root node is likely to be placed near the prior mean since the nodes with different depths will be placed at different levels of norms in the space.