# OpenReview forum: "A Rotated Hyperbolic Wrapped Normal Distribution for Hierarchical Representation Learning"
_NeurIPS.cc/2022/Conference — NeurIPS 2022 Accept_

### Official Review · Reviewer_nB4o · 2022-06-30

**Rating:** 6
**Confidence:** 4
**Soundness:** 3 good
**Presentation:** 4 excellent
**Contribution:** 3 good

**Summary:**

The authors build upon the Hyperbolic Wrapped Normal (HWN) distribution, which puts a distribution on hyperbolic space by transporting a distributed tangent vector from an origin to a mean point mu and taking the exponential map (a diffeomorphism in this case). The authors observe that a standard basis vector in the tangent plane at the origin, transported to point mu and then exp-mapped and projected to a Poincaré disk, is aligned with a standard basis vector in the disk. They posit that the resulting distribution is less desirable than a distribution that is aligned such that one of the tangent vectors points in the disk from mu to the origin. To fix this, they propose the Rotated HWN distribution that is constructed so that the first dimension of the tangent plane points from mu to the origin.

Empirically, the authors show that the RHWN outperforms HWN with diagonal and full covariance matrix on a synthetic task, a word embedding task and a task of embedding Atari rollouts. In the synthetic task, the goal is to have the hamming distance on a binary tree correlate with the (I presume geodesic) distance in embedding space or to correlate tree depth with the "Poincaré norm" (geodesic distance to origin, I presume). In the word embedding task, the goal is to have hypernymy correlate with distance. In the Atari task, the goal is to have distance to the origin correspond with progress in the game.

**Questions:**

Please see my point above on the isometry symmetries.

**Limitations:**

The authors fairly address the limitations of their work.
I disagree though that rotating the tangent space to the root node is applicable only on hyperbolic space, see my comment above.

**Strengths And Weaknesses:**

## Strengths:
- The paper is very clearly and pleasantly written with many insightful illustrations - critical when dealing with geometry.
- I like the construction of the paper: diagnosing an issue with a prior method, then proposing a simple fix
- The proposed method is simple to describe and implement.
- The proposed method outperforms prior work.

## Weaknesses
I have one major concern with the geometric analysis and some other points.
### Symmetry of the geometry
I am not sure that I agree with the diagnosis. As a geometric method, it shouldn't matter which model of the hyperbolic space we pick. Up to isometry there is only one space of n dimensions with constant -1 curvature. This space, $H_n$, is a homogenous (even symmetric) space meaning that there is an isometry mapping any tangent vector $v$ at point $p$ to any other tangent vector $v'$ at any point $q$ assuming that they have the same norm: $g_p(v, v)=g_q(v', v')$. Thus, I don't really see why in Fig 2 the unrotated image on the top and the rotated image on the bottom are different. I suspect there is an isometry that maps between them. The only reason the top image looks "odd" is the particular choice of coordinates. Similarly, the paper makes repeated reference to the "origin" and "standard basis" of the Poincaré disk, which are also contingent on a particular choice of coordinates. A geometric method shouldn't care about those. Hence, when constructing a single distribution in hyperbolic space, up to isometry, the added rotation in the RHWN effectively does nothing.

The main reason the isometry-symmetry is broken is the particular choice of prior $p(z)$ in the variational inference, taken to be centered around a particular point $\mu_0$, which can then be called the origin. I suppose that's what the authors do. This would be the only reason why the root of a tree should be embedded close to the origin - geometry itself doesn't promote that. Yet the authors make no mention of the prior $p(z)$ in their analysis. I suppose the RHWN works well because first the prior encourages the root to be close to the origin and then it being preferable to have the basis such that one dimension points towards the root.

I feel the analysis can be much improved if these considerations were taken into account. Concretely:
- I'd like to see an argument why the root node should be place at the mean of the prior $p(z)$. For that, I'd also like to know what's special about the root node. If one interprets a (e.g. binary) tree as a graph, then it can be seen as a binary tree with any node as root node. I'd guess that this has to do with the balancing of the resulting tree - or with some notion of distance between that the nodes that's not encoded in an (unweighted) tree.
- Then in the subsequent discussion, it'd be better if references to "standard basis" and "origin" (arbitrary coordinatization) would be made geometrically meaningful relative to the prior origin / placement of the root node.

### Other points
1. Due to my above points, it's not quite clear to me why the $x$ vector in the algorithm should be fixed, so that the first dimension of $v$ is rotated to point to the arbitrary origin. It would be interesting to have it be learned and given by the encoder network. This is somewhat similar to e.g. Householder flows [1]. I'd be great if the authors could compare to such a method. One motivation could be that one wants to align not to the root node, but to the parent node, which may not lie in the direction of the root node. It'd be very interesting if the learned point to rotate to would indeed be the parent.
2. Why would the rotation be particular to a hyperbolic embedding of trees? In other manifolds with a prior with a well-defined mean and exp-mapped normal distributions, one could rotate one tangent vector towards the prior mean. For Euclidean spaces, this would be straightforward, but I could also imagine this working on Lie groups [2]. I'd be very interested to see if that improves the performance. The paper could make a much more general claim if they show improvement across different kinds of manifolds.
3. The rotation matrix (4) is not the only rotation that maps $x$ to $y$. It'd be good if the authors discuss this and say why this particular one (which I believe is the one with the smallest SO(n) distance to the unit) is desirable.
4. The authors use the word "prior" confusingly to not as the distribution $p(z)$ in variational inference, but as the variational family the posterior $q(z|x)$ belongs to. I see how this originates from the idea of "prior" as domain knowledge, but I'd prefer a different wording for that.
5. Due to its simplicity, the proposed method is a quite incremental change over prior work, hence somewhat limited in originality.


# Conclusion
Because I don't think the current analysis is correct without explicit mention of the role of the prior, I propose rejection of the paper. If the authors address those issues satisfactorily, I would recommend acceptance. If the authors could expand the paper by addressing my points (1) and (2), this could become a strong paper.

-- Score updated
-- Score updated again

[1] Tomczak, Jakub M., and Max Welling. "Improving variational auto-encoders using householder flow." (2016).

[2] Falorsi, Luca, et al. "Reparameterizing distributions on lie groups." (2019)

---

> ### Author Response · Authors · 2022-08-02
> **Response to Reviewer nB4o (3)**
>
> **References**
>
> [1] Nickel, Maximillian, and Douwe Kiela. "Poincaré embeddings for learning hierarchical representations." Advances in neural information processing systems 30 (2017).
>
> [2] Nickel, Maximillian, and Douwe Kiela. "Learning continuous hierarchies in the lorentz model of hyperbolic geometry." International Conference on Machine Learning. PMLR, 2018.
>
> [3] Rik Sarkar. 2011. Low distortion delaunay embedding of trees in hyperbolic plane. In Proceedings of the 19th international conference on Graph Drawing (GD'11). Springer-Verlag, Berlin, Heidelberg, 355–366.
>
> [4] Wallach, Hanna, David Mimno, and Andrew McCallum. "Rethinking LDA: Why priors matter." Advances in neural information processing systems 22 (2009).
>
> [5] Ines Chami, Adva Wolf, Da-Cheng Juan, Frederic Sala, Sujith Ravi, and Christopher Ré. 2020. Low-Dimensional Hyperbolic Knowledge Graph Embeddings. In Proceedings of the 58th Annual Meeting of the Association for Computational Linguistics, pages 6901–6914, Online. Association for Computational Linguistics.
>
> [6] Y. Shavitt and T. Tankel, "Hyperbolic Embedding of Internet Graph for Distance Estimation and Overlay Construction," in IEEE/ACM Transactions on Networking, vol. 16, no. 1, pp. 25-36, Feb. 2008, doi: 10.1109/TNET.2007.899021.
>
> [7] Cid, J. Ángel, and F. Adrián F. Tojo. "A Lipschitz condition along a transversal foliation implies local uniqueness for ODEs." arXiv preprint arXiv:1801.01724 (2018).

---

> ### Author Response · Authors · 2022-08-02
> **Response to Reviewer nB4o (2)**
>
> **Due to my above points, it's not quite clear to me why the $x$ vector in the algorithm should be fixed, so that the first dimension of $v$ is rotated to point to the arbitrary origin. It would be interesting to have it be learned and given by the encoder network. One motivation could be that one wants to align not to the root node, but to the parent node, which may not lie in the direction of the root node. It'd be very interesting if the learned point to rotate to would indeed be the parent.**
>
> We conduct additional experiments with the learnable rotation model which learns the rotation direction $y$ in Algorithm 1. We test two variants of learnable rotation models on the noisy synthetic binary tree. Originally, the encoder of VAE outputs $\mu$ and $\Sigma$ of the variational distribution. With the first variant “Learnable Rotation 1”, the encoder additionally outputs the rotation direction. With the second variant “Learnable Rotation 2”, the output $\mu$ of the encoder is fed to another fully connected layer to predict the rotation direction.
>
> As shown in Table 8 (and below), as the depth becomes greater, the learnable rotation models usually underperform RoWN. Figure 6 in Appendix D.2 shows the behaviors of the learned representation by an alternative Learnable Rotation 1. Most of the rotation directions are pointing or orthogonal (black line segments on each node) to the direction of its parent node. This implies that the alternatives of RoWN can learn representations of nodes to align not to the root node but to the parent node. We note that these alternatives work well with shallow depths, but not with great depths.
>
> | **Depth** | **4** | **5** | **6** | **7** |
> |---|---|---|---|---|
> |**Correlation w/ distance**|
> | **RoWN** | $0.820_{\pm .015}$ | $0.807_{\pm .017}$ | $0.822_{\pm .017}$ | $0.788_{\pm .016}$ |
> | **Learnable Rotation 1** | $0.820_{\pm .013}$ | $0.804_{\pm .016}$ | $0.808_{\pm .013}$ | $0.770_{\pm .028}$ |
> | **Learnable Rotation 2** | $0.817_{\pm .018}$ | $0.811_{\pm .018}$ | $0.801_{\pm .008}$ | $0.772_{\pm .021}$ |
> |**Correlation w/ depth**|
> | **RoWN** | $0.930_{\pm .026}$ | $0.911_{\pm .027}$ | $0.901_{\pm .034}$ | $0.827_{\pm .047}$ |
> | **Learnable Rotation 1** | $0.952_{\pm .018}$ | $0.903_{\pm .037}$ | $0.845_{\pm .057}$ | $0.711_{\pm .072}$ |
> | **Learnable Rotation 2** | $0.948_{\pm .015}$ | $0.907_{\pm .019}$ | $0.844_{\pm .013}$ | $0.724_{\pm .058}$ |
>
>
> **Why would the rotation be particular to a hyperbolic embedding of trees? In other manifolds with a prior with a well-defined mean and exp-mapped normal distributions, one could rotate one tangent vector towards the prior mean. For Euclidean spaces, this would be straightforward, but I could also imagine this working on Lie groups [2]. I'd be very interested to see if that improves the performance. The paper could make a much more general claim if they show improvement across different kinds of manifolds.**
>
> We thank you for the insightful comment. We believe that the idea of RoWN is applicable to other Riemannian manifolds, where the closed-form of the basic operations, i.e. exponential mapping and parallel transport, are well understood.
>
> **The rotation matrix (4) is not the only rotation that maps $x$ to $y$. It'd be good if the authors discuss this and say why this particular one (which I believe is the one with the smallest $\operatorname{SO}(n)$ distance to the unit) is desirable.**
>
> Although there are multiple ways to rotate, in terms of likelihood or ELBO, the results will be the same whatever the method we use, as long as the method is used consistently. In this work, we choose the most commonly used rotation method [7].
>
> **The authors use the word "prior" confusingly to not as the distribution $p(z)$ in variational inference, but as the variational family the posterior $q(z \mid x)$ belongs to. I see how this originates from the idea of "prior" as domain knowledge, but I'd prefer a different wording for that.**
>
> We will clarify the prior and variational distributions appropriately in the revised manuscript. Thank you for the suggestion.
>
> **Due to its simplicity, the proposed method is a quite incremental change over prior work, hence somewhat limited in originality.**
>
> The suggested method may seem incremental at first glance, however, the proper choice of the distribution has been an important issue in probabilistic modeling, especially in the Bayesian methods. For example, one could think [4] as incremental work that contains an extensive comparison between asymmetric and symmetric Dirichlet distributions. However, due to their work, people can recognize the importance of asymmetric prior in that literature. We believe our work helps researchers understand the hyperbolic space better than before.

---

> > ### Comment · Reviewer_nB4o · 2022-08-08
> > **Thanks for the additional evaluations, updated score**
> >
> > I thank the authors for the additional experiments.
> >
> > Taking into account the other reviews and the rebuttals, I have increased my score to a borderline reject, because:
> > - I'm still not convinced of the soundness of the analysis and the revised version did not incorporate any notion of the role of the VAE prior.
> > - Regarding the root placement word-embedding argument, although I have doubts, I acknowledge that a similar analysis has been published before [1], so it makes sense for this work to build on top of that.
> > - The method is simple and yields a small improvement in performance, thus may be interesting to the community.

---

> ### Author Response · Authors · 2022-08-02
> **Response to Reviewer nB4o (1)**
>
> We sincerely thank you for the careful and insightful comments. We address your questions about the (i) symmetry of the geometry and (ii) other points.
>
> **Symmetry of the geometry**
>
> We agree that we assume the origin and the standard basis of the hyperbolic space are referenced based on the mean of the prior distribution (in VAE) throughout the manuscript. We implicitly made this assumption since the zero mean distribution will eventually be used as a prior in the models such as hyperbolic VAE, and we will clarify this point in the revised manuscript.
>
> Regarding the placement of the root in the hyperbolic space, we need to consider two different cases: probabilistic word embedding (Eq 5) and hyperbolic VAE.
>
> When RoWN is used as a distribution for probabilistic word embedding (Eq 5), the mean of the root node is not necessarily located near the origin if the structure of the dataset is arbitrary.
> Having said that, previous studies have shown that when one embeds an arbitrary tree into the hyperbolic space to maximally preserve the $\ell 2$ distances between nodes, the node with the highest centrality is likely to be placed near the origin due to the exponentially expanding property of the hyperbolic space [1]. With the same intuition, when a tree-structured dataset is embedded by using Eq 5, the root node is likely to be embedded near the origin since the root node is likely to have the highest centrality in the tree (and the KL divergence contains $\ell 2$ distances between means of two distributions).
>
> When RoWN is used as a variational distribution in the hyperbolic VAE, the situation is slightly different since the KL divergence encourages all variational means to be close to the prior mean, and we do not measure the pairwise difference between nodes. Having said that, if we only focus on nodes at a certain depth in a tree, it can be easily identified that it would be beneficial to have all nodes at the same level of the norm to minimize the geometric mean between the prior and posterior means (here we assume that each pair of nodes requires to have a certain amount of distance to minimize the reconstruction error). As we can observe from Figure 3a, since the original HWN cannot model the local variation along the radial direction, it is difficult to have the nodes at the same depth with similar norms. Eventually, the root node slightly deviates from the prior mean. With RoWN, as we can observe from Figure 3c, all the nodes at the same depth can be placed at a similar norm while preserving their local variations. If this is indeed the case, the root node is likely to be placed near the prior mean since the nodes with different depths will be placed at a different level of norms in the space.
>
> Of course, this is not always the case. If a dataset has a highly unbalanced tree structure, then the root node may not be placed near the origin. Having said that, most of the previous work on hyperbolic embedding focuses on balanced implicit/explicit hierarchical datasets, such as vocabulary with relationships [1, 2], knowledge graphs [5], and social networks [6] and shows the root node is embedded near the origin.

---

> > ### Comment · Reviewer_nB4o · 2022-08-08
> > **Analysis unchanged in revised version - still not convinced**
> >
> > ### Word embeddings
> > In the case of word embeddings, in which there is no prior, I expect loss (5) to be invariant to moving around all embeddings with an isometry. In hyperbolic space, an isometry exists between any two points. Also, the volume of all balls depends only on the radius, not on its centre point. I thus don't see an argument for why the root should be close to the centre.
> > I suspect that something in the neural parametrization makes that it is convenient for the network to place the root in the centre. Such an analysis would be more convincing than the current argument.
> > Thus, I am quite unconvinced why in this case, the RHWN method makes sense.
> > That said, I acknowledge that [1] does also make the current claim regarding root placement.
> >
> > > $l2$ distance
> >
> > Do you mean Euclidean distance or distance in hyperbolic space? The $l2$ norm generally refers to the former, but I don't see why that would make sense here.
> >
> > > when a tree-structured dataset is embedded by using Eq 5, the root node is likely to be embedded near the origin since the root node is likely to have the highest centrality in the tree (and the KL divergence contains $l2$ distances between means of two distributions).
> >
> > Could you point me to a reference for an expression of the KL divergence for your class of distributions in hyperbolic space that includes a $l2$ between the means? E.g. I can't find it in [8].
> >
> > ### Hyperbolic VAE
> > For the hyperbolic VAE, I see how it makes sense for a root node to be placed in the center due to the prior, which breaks the symmetry. However, I didn't see any changes in the revised version following my suggestions. I still see no mention of the VAE prior.
> >
> > > it can be easily identified that it would be beneficial to have all nodes at the same level of the norm to minimize the geometric mean between the prior and posterior means (here we assume that each pair of nodes requires to have a certain amount of distance to minimize the reconstruction error)
> >
> > Such an argument (perhaps made slightly more precise) should be in the paper to motivate why the origin is relevant.
> >
> > [8] Mathieu, Le Lan, Maddison, Tomioka, and Teh. n.d. “Continuous Hierarchical Representations with Poincaré Variational Auto-Encoders.” Advances in Neural Information Processing

---

> > > ### Author Response · Authors · 2022-08-09
> > > **Re-Response to Reviewer nB4o (2)**
> > >
> > > In the following, we will clarify a few mistakes in our original rebuttal.
> > >
> > > > Do you mean Euclidean distance or distance in hyperbolic space? The $\ell_2$ norm generally refers to the former, but I don't see why that would make sense here.
> > >
> > > Sorry for the confusion. The $\ell2$ used in “to maximally preserve the $\ell2$ distances between nodes” means the path length between the two nodes in the tree.
> > > The $\ell2$ used in “(and the KL divergence contains $\ell2$ distances between means of two distributions)” means the hyperbolic distance between the means.
> > >
> > > > Could you point me to a reference for an expression of the KL divergence for your class of distributions in hyperbolic space that includes a $\ell2$ between the means? E.g. I can't find it in [8].
> > >
> > > It is our mistake. Indeed there is no closed-form of the KL divergence between the HWN. Instead, we conduct another line of analysis to show that the KL divergence implies the hyperbolic distance between the means.
> > >
> > > To ensure the probabilistic hyperbolic embedding model behaves similar to the deterministic hyperbolic embedding model, i.e. the root node places near the origin, we use the Poincar\'e ball model, which is the isometry of the Lorentz model and has the closed-form density function of HWN [8].
> > > Let $d_p(x, y)$ be the distance function of the Poincar\'e ball, where $x, y$ are the points on the Poincar\'e ball.
> > > The log-density function of the wrapped normal distribution $p(z \mid \mu, \Sigma)$ in the Poincar\'e ball is computed as:
> > >
> > > \begin{align}
> > >     \log p(z \mid \mu, \Sigma) &= (k - 1) \log \frac{d_p(\mu, z)}{\sinh{d_p(\mu, z)}} + (\textrm{log probability of } \, \lambda_\mu \exp^{-1}_\mu(z) \, \textrm{from} \, \mathcal{N}(0, \Sigma)) \\\\
> > >     &\approx -(k - 1)\frac{d_p^2(\mu, z)}{6} + (\Sigma\textrm{ related term}),
> > > \end{align}
> > >
> > > where $k$ is the dimension of the latent space and Taylor series of $\log(x / \sinh(x)) \approx -x^2 / 6$ is used in the approximation.
> > >
> > > Now, given $s \sim t$ and $s \not\sim t’$, representing the presence and absence of hypernymy relation respectively, we can write the objective of the probabilistic word embedding model as:
> > >
> > > \begin{align*}
> > >     KL(q_s \parallel q_t) - KL(q_s \parallel q_{t'}) &= E_{z \sim q_s(z)} \left[ \log \frac{q_s(z)}{q_t(z)} - \log \frac{q_s(z)}{q_{t'}(z)} \right] \\\\
> > >     &\approx \frac{k - 1}{6} E_{z \sim q_s(z)} \left[ d_p^2(\mu_t, z) - d_p^2(\mu_{t'}, z) + (\Sigma\textrm{ related terms})\right],
> > > \end{align*}
> > >
> > > which implies that minimizing the objective is equivalent to making $\mu_t, \mu_s$ closer than $\mu_{t'}, \mu_s$.

---

> > > > ### Comment · Reviewer_nB4o · 2022-08-09
> > > > **Thank you - agreed**
> > > >
> > > > Thank you very much for the new analysis and the experiments ran at such short notice. I agree with your current analysis.
> > > > I've updated my score to a weak accept as I no longer have concerns regarding soundness and think the paper has a moderate-to-high impact.

---

> > > ### Author Response · Authors · 2022-08-09
> > > **Re-Response to Reviewer nB4o (1)**
> > >
> > > Thank you for the additional comments and suggestions. We added the discussions about the symmetry of the geometry in Appendix D of the revised manuscript with an indicator in the last paragraph of Section 4.3. Due to the short time limit of rebuttal, we have added the additional results and discussions in the appendix. In the future revision, we will formalize the discussion and revise the main text to make the manuscript comprehensive.
> > >
> > > > Placement of root under isometry of geometry.
> > >
> > > (Thank you for the patience and for providing further details to clarify the comment. Our initial rebuttal did not address the comment properly) We agree that due to the isometry, the root node is not necessarily located near the origin. In fact, the analysis in [1] seems more about the possibility of lossless embedding of tree-structured data in the hyperbolic space without careful consideration of the root placement.
> > >
> > > We have further analyzed why in our case the root is placed near the origin and found that the initialization matters in this case. In our experiments, we have initialized embeddings near zero vectors, which are then moved to the Lorentz model using the exponential map, for the numerical stability of the optimization process (as suggested in [1]), which breaks the symmetry of the geometry we think. Although the initialization bias is not strong as the prior in VAE, in our additional experiments, we have seen that without zero initialization the root can be placed anywhere while having the same level of MR and mAP. See the table below for the detailed experimental results (‘near zero vector’ initializes the embeddings close to zero vector, and ‘near one vector’ initializes the embeddings close to one vector).
> > >
> > > | **Latent dimension** | **2** | **5** | **10** | **20** |
> > > |---|---|---|---|---|
> > > |**MR**|
> > > | **Near zero vector** | $4.346_{\pm .643}$ | $3.270_{\pm .144}$ | $2.828_{\pm .098}$ | $2.508_{\pm .049}$ |
> > > | **Near one vector** | $4.029_{\pm .415}$ | $3.209_{\pm .094}$ | $2.856_{\pm .075}$ | $2.491_{\pm .051}$|
> > > |**mAP**|
> > > | **Near zero vector** | $0.821_{\pm .016}$ | $0.891_{\pm .006}$ | $0.891_{\pm .005}$ | $0.895_{\pm .003}$ |
> > > | **Near one vector** | $0.829_{\pm .010}$ | $0.894_{\pm .004}$ | $0.891_{\pm .004}$ | $0.896_{\pm .003}$|
> > > |**Poincar\'e norm of the root node**|
> > > | **Near zero vector** | $0.122_{\pm .036}$ | $0.042_{\pm .015}$ | $0.031_{\pm .007}$ | $0.024_{\pm .004}$ |
> > > | **Near one vector** | $0.505_{\pm .076}$ | $0.650_{\pm .010}$ |$0.732_{\pm .003}$ | $0.801_{\pm .001}$ |
> > >
> > > Similarly, we conjecture that a similar effect can be shown with explicit weight decaying, which will be explored further in the future. We will clarify this point in our future revision. Again, thank you for your careful analysis of this matter.

---

### Official Review · Reviewer_hPAy · 2022-07-08

**Rating:** 7
**Confidence:** 4
**Soundness:** 4 excellent
**Presentation:** 3 good
**Contribution:** 3 good

**Summary:**

In this submission the authors introduce a parametric family of probability distribution on the hyperbolic space. They build on the hyperbolic wrapped normal distribution which rely on pushing Gaussian samples with parallel transport to the mean and then along the exponential map onto the manifold. Their main insight is based on the fact that when using a diagonal covariance matrix, the arbitrary choice of basis to sample this Gaussian (in the tangent space at the origin) is carried to the tangent space at the mean. Thus the principles axes of the distribution would not (apart if the mean lies on the x or y axes) be aligned with the "hierarchical depth direction" (which is given by the mean) and the "hierarchical same-level direction) which is the hyperplane orthogonal to the "depth" direction. They highlight that these directions are the "natural" ones over which the uncertainty/variation should be expressed, and therefore proposed to rotate the diagonal covariance matrix so that these are aligned.
They empirically demonstrate that this parametric distribution is better able than the diagonal and full covariance variants in learning latent representation that are better correlated with variables of interest, hence leading to "better" learnt hierarchical representations in that sense.


**Questions:**


Global comments:
- I would strongly suggest moving the figures/tables so that they are closer (in the same page) as the related paragraph. At the moment it's not the case, and I believe it would greatly improve the reader's experience.
- As the proposed method is de facto a specific setting of the full covariance matrix, I believe that the investigation of the failure of full covariance model would be worthy of more space. I believe that Figure 3 is useful in that regard but I have a few remarks/questions on it later. It is claimed that this parametrisation lead to unstable optimisation, but it would be nice to dive a bit more, e.g. computing the condition number. Additionally, the isotropic variant where the covariance matrix is given by $\Sigma = \sigma I_d$ is not explored, whilst it would be easy to do so and interesting to see whether it is indeed not flexible enough.


Specific comments:
- typo "Hyperboilc" in openreview title
- Abstract: "RoWN, the newly proposed distribution" -> "newly" perhaps misleading as it's in some ways a specific setting of the full covariance matrix (which now depends on the mean).
- Section 2
    - Sec 2.3: I feel that the sampling process for the HWN would really benefit from having an illustrative figure on the side. This could also be used to illustrate Algorithm 1, which is the core contribution of the paper.
- Section 3
    - Sec 3.1: Proposition 1 is somewhat repeated before *and* after which seems unnecessary.
    - Sec 3.2:
        - Worth noting in Algorithm 1, that to sample the multivariate normal $v \sim \mathcal{N}(0, \hat{\Sigma})$, one can rely on computing $\hat{\Sigma}^{1/2} = R \text{diag}(\sqrt{\sigma})$ (which exist and is unique since $\Sigma$ is positive definite) which can be done easily.
        Figure 3: "different prior choices" -> meant "posterior"? What are the dashed blue lines? Although I do find the upper row visually pretty, I don't understand from which posterior distributions the samples come from (from the ones whose means are represented in red?) nor the signification of the colour.
- Section 4
    - Sec 4.1:
        - I would suggest giving a bit more of background on the model for the noisy synthetic tree, mainly on the loss and likelihood (based on the Hamming distance) and how this plays with the spherical noise since the nodes aren't binary anymore.
        - Also I would be curious to see Table 1 for <4 and >7 depth, are the results similar?
        - Additionally I was wondering whether "better" latent representations (measured via the reported correlations) lead to better likelihood (as estimated via IWAE [1])?
    - Sec 4.2:
        - The HWN (full covariance) dramatically fails, according to the author this due to "unstable optimization". What is the reason why this phenomenon did not occur in the VAE setting from Section 4.1?
        - Table 2: Would suggest adding "metric" and "model" in the table's header
        - Worth reminding what are the MR and mAP metrics for the readers unfamiliar with these? either via their mathematical formula or in an english sentence.
    - Sec 4.3:
        - What is the motivation for the Poincaré norm of the latent representation to be correlated with the score? I'm only partially familiar with breakout, is the score correlated with the number of broken blocks? Isn't the Poincaré norm simply correlated with time then?
- Sec 5:
    - The reader may benefit from starting the related work section with a paragraph on "hierarchical representation learning". These methods are most of the time relying on discrete hierarchical structures, there are quite a few in the Bayesian non-parametrics literature (e.g. [2, 3]).
    - In the "Distributions in the hyperbolic space" paragraph, one could also mention normalizing flow related methods (e.g. [4, 5]).


**Limitations:**

I believe so

**Strengths And Weaknesses:**


The proposed idea is simple, pretty well presented and well executed, with nice figures to illustrate the model and pretty convincing empirical results.
The paper was mostly pleasant to read and easy to follow.

I don't see any major weakness.
In the following section I make suggestions to improve the organisation of the paper, along with some questions which I hope will lead to some clarification.

---

> ### Author Response · Authors · 2022-08-02
> **Response to Reviewer hPAy (2)**
>
> **Sec.4.1-2) Also I would be curious to see Table 1 for <4 and >7 depth, are the results similar?**
>
> We additionally conduct experiments for depth 3 and 8. For depth 3, the results are similar to the other depths as seen in the table below.
>
> For depth 8, NaN appeared in the hyperbolic models so we failed to obtain the full results. We analyze this is because extremely high values appear in the encoder outputs due to the $\cosh$ function in the exponential map. Clipping the values carefully may be helpful for training the hyperbolic models.
>
> | **Models** | **Euclidean** | **HWN (diagonal $\Sigma$)** | **HWN (full $\Sigma$)** | **RoWN** |
> |---|---|---|---|---|
> |Correlation w/ distance | $0.803_{\pm 0.037}$ | $0.844_{\pm 0.021}$ | $0.833_{\pm 0.029}$ | $0.839_{\pm 0.026}$ |
> |Correlation w/ depth | $0.882_{\pm 0.047}$ | $0.957_{\pm 0.013}$ | $0.938_{\pm 0.062}$ | $0.970_{\pm 0.011}$ |
>
> **Sec.4.1-3) Additionally I was wondering whether "better" latent representations (measured via the reported correlations) lead to better likelihood (as estimated via IWAE [1])?**
>
> In Appendix D, we report the ELBOs of the experiments. The results show that the ELBOs are similar across different distributions.
>
> ELBO is inappropriate to measure the representations due to the identifiability problem [2], such that different representations can have the same lower bound. In Appendix D, Table 7 and 14 show the ELBOs of the experiments with different prior and variational distributions. The ELBO scores are similar although their correlations can vary a lot.
>
> **Sec.4.2-1) The HWN (full covariance) dramatically fails, according to the author this due to "unstable optimization". What is the reason why this phenomenon did not occur in the VAE setting from Section 4.1?**
>
> We conduct an additional analysis on what causes the optimization instability and find that the number of samples used to approximate the KL divergence is critical to full covariance HWN, especially in the Wordnet dataset. In VAE, we can observe more stable results. We speculate that the stability improved since the relatively simple prior (standard normal distribution) is employed with the full covariance variational distribution.
>
> The additional results are provided in Table 11 (and below). As the number of samples increases, the performance increases but the result is still poor, and using more samples leads to higher time complexity.
>
> | **# of samples** | **1** | **50** | **100** |
> |---|---|---|---|
> | MR | $326.185_{\pm 7.610}$ | $73.572_{\pm 9.926}$ | $69.964_{\pm 10.457}$ |
> | mAP | $0.056_{\pm 0.004}$ | $0.241_{\pm 0.021}$ | $0.241_{\pm 0.019}$ |
> | Runtime (s/epoch) | 6.288 | 6.748 | 8.163 |
>
> **Sec.4.3) What is the motivation for the Poincaré norm of the latent representation to be correlated with the score? I'm only partially familiar with breakout, is the score correlated with the number of broken blocks? Isn't the Poincaré norm simply correlated with time then?**
>
> The scores are earned from the number of broken blocks, and this leads to the correlation between Poincar\’e norm and the elapsed time. The motivation of the Atari 2600 Breakout experiment is that the trajectories of the game are represented as a tree, where the depth is the number of broken blocks [1].
>
> **Sec.5) Advices for the related work**
>
> We thank you for the advice about the related work. However, more detailed information on the references introduced in the reviews is missing (only the numbers exist). We hope the detailed information on the references to be added in future comments, so we can add them to the revised version.
>
> **References**
>
> [1] Nagano, Yoshihiro, et al. "A wrapped normal distribution on hyperbolic space for gradient-based learning." International Conference on Machine Learning. PMLR, 2019.

---

> ### Author Response · Authors · 2022-08-02
> **Response to Reviewer hPAy (1)**
>
> We acknowledge your help for improving our paper with careful reviews and comments. We will revise the manuscript based on all editorial comments including typo in the title. We address the specific comments and concerns in detail below.
>
> **Additionally, the isotropic variant where the covariance matrix is given by $\Sigma = \sigma I_d$ is not explored, whilst it would be easy to do so and interesting to see whether it is indeed not flexible enough.**
>
> We first emphasize that the isotropic covariance matrix remains the same after rotating (i.e., $\sigma I_d = \sigma R R^T = \sigma R I_d R^T$). The results with isotropic distributions are added in Appendix D. In the experiments, the isotropic HWN sometimes performs better than the other models in noisy synthetic binary tree with respect to the correlation with distance, and WordNet with respect to the mean rank and mean average precision. In Atari 2600 Breakout experiment, the isotropic HWN shows better hierarchical representations than the other while showing relatively worse ELBO values.
>
> **Sec 2.3) I feel that the sampling process for the HWN would really benefit from having an illustrative figure on the side. This could also be used to illustrate Algorithm 1, which is the core contribution of the paper.**
>
> We thank you for the suggestion and agree that visualization for the sampling process will enhance the reader experience. We will add figures in the revisions.
>
> **Sec.3.2) Figure 3: "different prior choices" $\rightarrow$ meant "posterior"? What are the dashed blue lines? Although I do find the upper row visually pretty, I don't understand from which posterior distributions the samples come from (from the ones whose means are represented in red?) nor the signification of the colour.**
>
> To clarify, the noisy synthetic binary tree used in experiment 1 has two discrete children for each node. In Figure 3, we use 20 discrete children made by partitioning $[0, 3/2 \pi]$ to 20 pieces. We then add spherical noise in range $[0, \pi / 20]$, which divides the data points generated from different discrete children.
>
> The color in the upper row represents the level of spherical noise of the sample and the blue dashed line in the below row represents the mean representations of the variational distributions. The small red dots are the representative discrete children: $\{ [1, 0, 1], [1, 0, -1], [1, 1, 0], [1, -1, 0] \}$.
>
> **Sec.4.1-1) I would suggest giving a bit more of background on the model for the noisy synthetic tree, mainly on the loss and likelihood (based on the Hamming distance) and how this plays with the spherical noise since the nodes aren't binary anymore.**
>
> Each data point of a noisy synthetic binary tree consists of real numbers in range $[0, 1]$ as shown in Figure 4a, which is the same as normalized images. So training VAE on the noisy synthetic binary tree is exactly the same as training VAE on normalized images. We don’t use the information of Hamming distances in the training phase.
>
> We train the VAE with the data points with spherical noises, where the original synthetic binary tree is unseen. Then we evaluate the learned representations using the original synthetic binary tree, where the Hamming distances are now available.

---

### Official Review · Reviewer_hLZa · 2022-07-11

**Rating:** 5
**Confidence:** 3
**Soundness:** 4 excellent
**Presentation:** 3 good
**Contribution:** 1 poor

**Summary:**

This paper studies the geometry of two sub-classes of the Hyperbolic Wrapped Normal distribution (HWN) with the goal deepen the understanding of which family is more appropriate for probabilistic machine learning when the data are in (or can be embedded into) hyperbolic spaces. The first subclass is the diagonal HWN that has been applied lately to different problems in hyperbolic probabilistic machine learning and the second the rotated HWN, that is proposed by the authors as a more appropriate alternative.

The diagonal HWN draws a sample in two steps. First samples from a diagonal Gaussian in the tangent space at the origin and secondly parallel transports the sample to the new mean and projects it to the manifold via the exponential map. The proposed rotated HWN interpolates an extra step. After sampling at the origin, it rotates the sample by the rotation matrix that brings the origin to the direction of the new mean and then performs last step of the diagonal HWN. The authors provide a proof (and an additional visual explanation) that for both the diagonal HWN and the rotated HWN the principal axes become geodesics that intersect orthogonally on the new mean. From that the authors infer that the rotated HWN is more appropriate (especially for the Poincare model) since the principal axes correspond to the radial and angular variations accordingly.

They perform 3 experiments on data that have hierarchical structure (and thus better embedded in hyperbolic space) to support their claims. The first is a toy dataset of a noisy binary synthetic tree. The goal is to show that the rotated HWN generrates posteriors that capture better the distances between the nodes as well as their level in the hierarchy (via their norm). In the second task a real language dataset is used and the goal is to embed the words and count their rank. From that task, the authors provide experimental proof that the full HWN is not appropriate for some hierarchical tasks and conjecture that this is due to the difficulty of the optimization problem. The last task is to embed the trees that can describe the evolution of an Atari game. The goal is to show that the angular direction in the rotated HWN is more informative than the one of the diagonal HWN, by describing "sibling states" of equivalent scores.

**Questions:**

1) In Algorithm 1 why do we need to resort to the inner product of the ambient space i.e. a)the normalization of $\mu$ is w.r.t. the Euclidean norm and not the norm induced by the Lorentz inner product, b) the rotation matrix.

2) The inner product $\langle \cdot,\cdot \rangle_x$ is defined but not used.

4) How are the correlation metrics in experiment 1 computed exactly and why are they more appropriate to present than the ELBO?

**Limitations:**

The authors discuss some limitations of the proposed distribution, i.e. that it only works for hyperbolic spaces (although it seems relevant to a lot of conformal maps) and that the full HWN might be better optimized if there were better optimization algorithms (although more samples might still be needed to distinguish the best distribution in the family)

-I would advise the authors to extend their idea to more general families of distributions in hyperbolic space as well as more general conformal maps.

-Moreover, to strengthen the intuition about how the angular and radial dependency decompose more appropriately the authors should provide the marginal distributions and an extensive discussion.

-(Minor) The authors should mention what exactly is the "hyperbolic VAE" and how it is modified by them in the main text.

**Strengths And Weaknesses:**

Originality:

(+) To my knowledge the observations and proofs in this paper are original.

(-) The figures describing the spaces and the experimental evaluation are not novel but they seem standard.

Quality:

(+) High quality in writing and presentation.

Clarity:

(+) Paper is clearly presented.

(-) (Minor) Figure 2 (a): "red" and "blue" are interchanged, since red is the major principal axis.

(-) In the preliminaries one more sentence about what is a geodesic would be appropriate.

Significance:

(-) the paper proposes a simple modification to constraint the family of HWN.

(+) However, this observation aligns well with the intuition gained from the conformality of the map from Euclidean to Poincare disk. It also disentangles better the angular and radial dimensions and the authors provide proofs, figures to build intuition and standard experimental evaluation.

---

> ### Author Response · Authors · 2022-08-02
> **Response to Reviewer hLZa (2)**
>
> **Moreover, to strengthen the intuition about how the angular and radial dependency decompose more appropriately the authors should provide the marginal distributions and an extensive discussion.**
>
> We compute the Pearson correlation between the radial axis and the angular axes in the noisy synthetic binary tree task, which has been shown an effective measure of variable dependency in disentangled representation learning [4, 5], to show how well the angular and radial dependencies are decomposed. Table 8 (and below) shows the correlation between the raidal axis and the angular axes. Through the experiments, we find that the absolute value of the correlation is lower or similar to 0.1 in all the models including RoWN.
>
> | **Depth** | **4** | **5** | **6** | **7** |
> |---|---|---|---|---|
> | **Correlation btw. $r$ and $\theta_1$** |
> | **Euclidean**               | $0.144_{\pm .170}$ | $0.007_{\pm .105}$ | $0.039_{\pm .106}$ | $-0.026_{\pm .093}$ |
> | **HWN (diagonal $\Sigma$)** | $0.025_{\pm .150}$ | $0.015_{\pm .137}$ | $0.110_{\pm .065}$ | $-0.003_{\pm .134}$ |
> | **HWN (full $\Sigma$)**     | $-0.053_{\pm .216}$ | $0.049_{\pm .136}$ | $-0.017_{\pm .169}$ | $0.012_{\pm .066}$ |
> | **RoWN**                    | $0.080_{\pm .163}$ | $0.030_{\pm .097}$ | $0.112_{\pm .093}$ | $0.025_{\pm .083}$ |
> | **Correlation btw. $r$ and $\theta_2$** |
> | **Euclidean** | $0.116_{\pm .232}$ | $-0.039_{\pm .210}$ | $0.109_{\pm .131}$ | $0.006_{\pm .095}$|
> | **HWN (diagonal $\Sigma$)** | $0.066_{\pm .170}$ | $-0.021_{\pm .190}$ | $0.044_{\pm .122}$ | $-0.024_{\pm .115}$|
> | **HWN (full $\Sigma$)** | $0.297_{\pm .116}$ | $0.113_{\pm .109}$ | $0.013_{\pm .122}$ | $0.061_{\pm .103}$|
> | **RoWN** | $0.013_{\pm .178}$ | $0.025_{\pm .173}$ | $-0.004_{\pm .115}$ | $0.064_{\pm .082}$|
> | **Correlation btw. $r$ and $\theta_3$** |
> | **Euclidean** | $0.067_{\pm .220}$ | $-0.110_{\pm .220}$ | $-0.016_{\pm .159}$ | $0.011_{\pm .098}$|
> | **HWN (diagonal $\Sigma$)** | $0.123_{\pm .252}$ | $-0.139_{\pm .123}$ | $-0.019_{\pm .133}$ | $-0.095_{\pm .101}$|
> | **HWN (full $\Sigma$)** | $-0.053_{\pm .144}$ | $-0.088_{\pm .107}$ | $0.106_{\pm .120}$ | $0.024_{\pm .079}$|
> | **RoWN** | $0.127_{\pm .253}$ | $-0.120_{\pm .169}$ | $-0.012_{\pm .120}$ | $-0.012_{\pm .083}$|
> | **Correlation btw. $r$ and $\theta_4$** |
> | **Euclidean** | - | $-0.015_{\pm .073}$ | $0.013_{\pm .081}$ | $0.026_{\pm .065}$|
> | **HWN (diagonal $\Sigma$)** | - | $-0.047_{\pm .117}$ | $-0.035_{\pm .147}$ | $0.080_{\pm .096}$|
> | **HWN (full $\Sigma$)** | - | $0.079_{\pm .150}$ | $0.042_{\pm .108}$ | $0.086_{\pm .102}$|
> | **RoWN** | - | $-0.031_{\pm .116}$ | $-0.070_{\pm .112}$ | $0.086_{\pm .115}$|
> | **Correlation btw. $r$ and $\theta_5$**|
> | **Euclidean** | - | - | $0.082_{\pm .075}$ | $-0.029_{\pm .109}$|
> | **HWN (diagonal $\Sigma$)** | - | - | $0.022_{\pm .101}$ | $-0.041_{\pm .097}$|
> | **HWN (full $\Sigma$)** | - | - | $-0.058_{\pm .111}$ | $0.076_{\pm .064}$|
> | **RoWN** | - | - | $-0.016_{\pm .111}$ | $-0.030_{\pm .112}$|
> | **Correlation btw. $r$ and $\theta_6$** |
> | **Euclidean** | - | - | - | $-0.018_{\pm .073}$|
> | **HWN (diagonal $\Sigma$)** | - | - | - | $-0.030_{\pm .082}$|
> | **HWN (full $\Sigma$)** | - | - | - | $0.035_{\pm .103}$|
> | **RoWN** | - | - | - | $0.006_{\pm .067}$|
>
> **(Minor) The authors should mention what exactly is the "hyperbolic VAE" and how it is modified by them in the main text.**
>
> Hyperbolic VAE is the VAE that uses the hyperbolic space as the latent space. The variational distribution of the hyperbolic VAE is modified to diagonal HWN, full covariance HWN, and RoWN.
> The additional text is added at Appendix E.1.
>
>
> **References**
>
> [1] Cid, J. Ángel, and F. Adrián F. Tojo. "A Lipschitz condition along a transversal foliation implies local uniqueness for ODEs." arXiv preprint arXiv:1801.01724 (2018).
>
> [2] Bishop, Christopher M. Pattern Recognition and Machine Learning. New York :Springer, 2006.
>
> [3] Nickel, Maximillian, and Douwe Kiela. "Poincaré embeddings for learning hierarchical representations." Advances in neural information processing systems 30 (2017).
>
> [4] J. Jo and J. Seo, "Disentangled Representation of Data Distributions in Scatterplots," 2019 IEEE Visualization Conference (VIS), 2019, pp. 136-140, doi: 10.1109/VISUAL.2019.8933670.
>
> [5] Horan, D., Richardson, E., & Weiss, Y. (2021). When Is Unsupervised Disentanglement Possible? In M. Ranzato, A. Beygelzimer, Y. Dauphin, P. S. Liang, & J. W. Vaughan (Eds.), Advances in Neural Information Processing Systems (Vol. 34, pp. 5150–5161). Curran Associates, Inc.
>
> [6] (2008). Pearson’s Correlation Coefficient. In: Kirch, W. (eds) Encyclopedia of Public Health. Springer, Dordrecht. https://doi.org/10.1007/978-1-4020-5614-7_2569

---

> ### Author Response · Authors · 2022-08-02
> **Response to Reviewer hLZa (1)**
>
> We appreciate your constructive feedback. Here, we address your questions and the limitations mentioned. All editorial comments (Clarity (-)’s, Question 2, etc.) will be addressed appropriately in the revised manuscript.
>
> **In Algorithm 1 why do we need to resort to the inner product of the ambient space i.e., a) the normalization of $\mu$ is w.r.t. the Euclidean norm and not the norm induced by the Lorentz inner product, b) the rotation matrix.**
>
> To rotate the diagonal HWN, we rotate the diagonal distribution ***defined on the tangent space*** in the direction of $\mu \in \mathbb{L}^n$. This is where we use the inner product of the ambient space. For the rotation, we need a vector that represents the direction of $\mu$ in the tangent space. Thanks to the property of the Poincar\’e disk model [3], where the angle between two vectors is the same as the Euclidean space and the projection function in Proposition 2, we know that $\mu_{1:}$ is the vector that we need. We use the normalized version of $\mu_{1:}$ since the rotation formula used in Algorithm 1 [1] takes two unit vectors as the inputs.
>
> **How are the correlation metrics in experiment 1 computed exactly and why are they more appropriate to present than the ELBO?**
>
> The correlation metrics are measured using Pearson correlation [6]. For example, the correlation between depth and Poincar\’e norm is measured by computing Pearson correlation between the list of depths and Poincar\’e norms.
>
> Especially, the ELBO may be inappropriate to measure the quality of the representations in some cases, where the identifiability problem arises, so different representations can provide the same values in terms of the ELBO, as addressed in [2]. We observe similar behaviors in our settings, too. The results for the ELBO values are available in Table 7 and 14.

---

### Author Response · Authors · 2022-08-02
**Special thanks to the reviewers**

We thank all reviewers for the insightful comments and suggestions about future work.
For each reviewer, we responded to the corresponding review.
We also uploaded the revised version of the manuscript and highlighted the changes with magenta colors.
Additional feedback and discussions are always welcome!

---

### Meta-Review · Area_Chair_rFXx · 2022-08-27

**Recommendation:** Accept
**Confidence:** Certain

**Metareview:**

The paper proposes the rotated hyperbolic wrapped normal (RoWN) distribution which improves hyperbolic wrapped normal (HWN) distribution in representing hierarchical (implicitly tree structured) datasets e.g., WordNet. The idea is to include a learnable rotation matrix in the generative process. The initial soundness concern raised by one of the reviewers was resolved in the discussion phase. This looks like a relevant contribution to the geometric deep learning community and I would like to recommend acceptance for this paper.

**Award:**

No

---

### Decision · Program_Chairs · 2022-09-14

Accept